# Tokenization to Transfer: Do Genomic Foundation Models Learn Good Representations?

**Kirill Vishniakov[1,2], Karthik Viswanathan[2], Aleksandr Medvedev[2],**
**Praveenkumar Kanithi[2], Marco AF Pimentel[2], Ronnie Rajan[2], Shadab Khan[2,3]***
[1]Ruya AI    [2]M42    [3]ADIA Lab

## Abstract

The success of Large Language Models has inspired the development of Genomic Foundation Models (GFMs) through similar pretraining techniques. However, the relationship between pretraining performance and effectiveness in downstream genomic tasks remains unclear. Additionally, the high computational cost of pretraining raises questions about its cost-efficiency. To assess the usefulness of pretraining in genomics, we evaluated seven different GFMs across 52 diverse genomic tasks, comparing them to their counterparts with randomly initialized weights. Across benchmarks, we find that randomly initialized models provide surprisingly strong baselines and tokenizer and architecture choices strongly shape both these baselines and the gains from pretraining. Specifically, character-token models often match or exceed the performance of larger pretrained k-mer or BPE models, whereas subword models appear to benefit from pretraining. We also find that the evaluated GFMs fail to capture clinically relevant genetic mutations, with embeddings and log-likelihood ratios showing limited sensitivity to annotated variants. For the tasks we study, these results suggest that current NLP-style pretraining strategies provide modest, tokenizer-gated improvements over strong random baselines and motivate more biologically informed tokenization and variant-aware objectives. Our code is available at github.com/m42-health/gfm-random-eval.

## 1 Introduction

Recent advances in language modeling have led to the application of similar unsupervised pretraining methods in genomics. This facilitated the emergence of Genomic Foundation Models (GFMs) (Consens et al., 2025) which learn representations from genomic sequences. This line of work has gained significant attention due to the potential of GFMs to revolutionize our understanding of genomics (Benegas et al., 2025b).

GFMs typically use a two-step training approach akin to Large Language Models: unsupervised pretraining on a large dataset, followed by a supervised finetuning on a downstream task. The pretraining phase usually involves either next token prediction (Brown et al., 2020) or masked language modeling (Devlin et al., 2019). The promise of unsupervised pretraining is to extract knowledge from vast genomic datasets (Consortium et al., 2015) and compress it into the model's parameters, with the aim of producing a generalist model applicable to a diverse set of tasks.

While some studies have explored scaling laws for GFMs (Nguyen et al., 2023; 2024), the relationship between pretraining and downstream performance remains unclear, with no single GFM consistently proving to be the best (Marin et al., 2024; Milovanović & Orvieto, 2025). Combined with large model sizes (Dalla-Torre et al., 2024), long input sequences (Nguyen et al., 2023; 2024; Brixi et al., 2025) and massive datasets, the pretraining step demands substantial computational resources.

The natural question arises: *how effective is unsupervised pretraining in the genomics domain?* To answer this, we conduct extensive experiments with seven recent GFMs across finetuning, feature extraction and genomic variation analysis as summarized in Fig. 2.

---

*S.K. was with M42 (initially) and ADIA Lab (subsequently) during this research work.

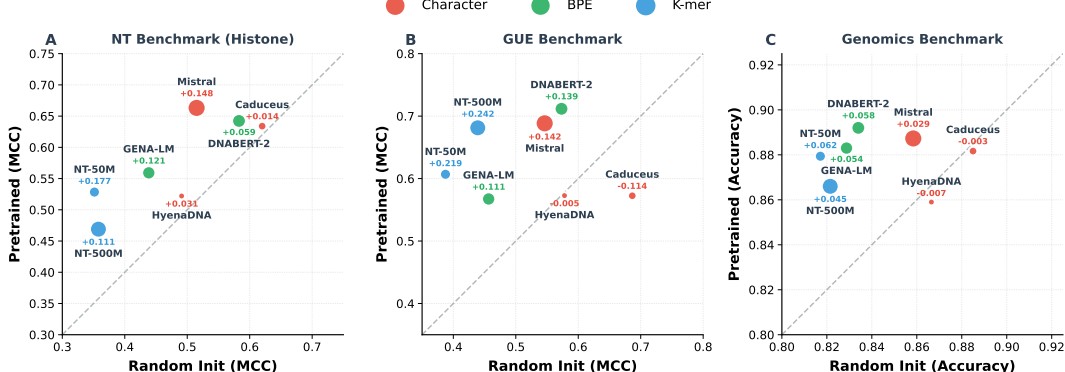

Figure 1: **Tokenizer choice determines random baseline quality, while pretraining gains are more complex.** A comparison of an averaged performance of identical models with random initialization ($x$-axis) versus pretrained ($y$-axis) across **(A)** NT Benchmark (12 histone and enhancer tasks), **(B)** GUE (7 task categories), and **(C)** Genomic Benchmarks (8 tasks). Points above the diagonal indicate a benefit from pretraining. **Finding:** The random baseline performance is strongly determined by the tokenizer, with character models (Orange) consistently outperforming subword models (Blue, Green). In contrast, the performance gains from pretraining are largest for subword models (NT-500M $\Delta + 0.242$ MCC on GUE) but are highly variable for character models. Marker size is scaled with the number of model parameters.

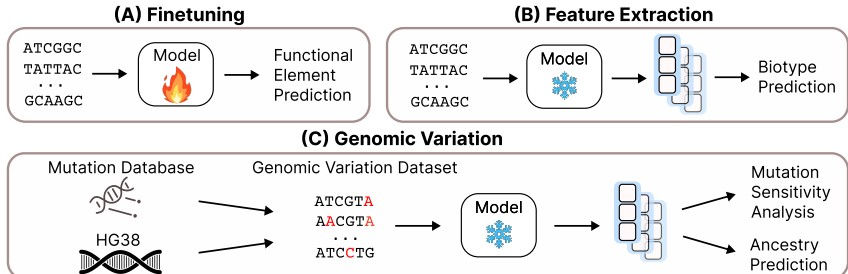

Figure 2: **Overview of the experiments.** (A) **Finetuning:** we finetune models on different functional element classification tasks. (B) **Feature Extraction:** For biotype classification, we extract embeddings from frozen models and train a simple classifier to predict gene types using these embeddings. (C) **Genomic Variation:** We evaluate models' ability to capture genetic variations through: (1) Mutation sensitivity analysis measures how well models distinguish between original and mutated sequences by computing embedding similarities, and (2) Ancestry prediction uses model embeddings with XGBoost to classify population groups based on genomic variants.

First, in standard finetuning tasks such as NT Benchmark (Dalla-Torre et al., 2024), GUE (Zhou et al., 2024), and Genomic Benchmarks (Grešová et al., 2023), we now compare each pretrained GFM directly to an identically configured model with random initialization (Fig. 1). Random baselines are surprisingly strong and depend strongly on tokenizer: character-level models (Caduceus, HyenaDNA, Mistral) achieve higher performance from scratch than models using k-mer or BPE tokenization. At the same time, pretraining yields consistent gains for k-mer and BPE models, while gains for character-token models are smaller and more architecture-dependent; a randomly initialized Caduceus model often matches or exceeds much larger pretrained GFMs across benchmarks.

This surprising trend continues to be observed in feature extraction task (Table 2), where embeddings from frozen models are used to train a simple classifier. One might expect the benefits of pretraining to be most pronounced here, because randomly initialized models receive no tuning whatsoever and their weights remain fully random. Instead, we find that randomly initialized models can be competitive, and that simple architectural choices, particularly using a character tokenizer and a larger embedding dimension, substantially improve their features. On the biotype classification task, an untrained HyenaDNA variant yields the best performance among the GFMs we evaluate, and a tokenizer ablation with matched HyenaDNA architectures (Table 3) shows that changing only

| Model | #Params | Architecture | Tokenizer | Vocab Size | Seq Len (tokens) | #Tokens | Data |
|-------|---------|--------------|-----------|------------|------------------|---------|------|
| HyenaDNA | 450K | Decoder | Char | 12 | 1024 | 2.6B | HRG |
| NT 500M | 500M | Encoder | k-mer | 4107 | 1000 | 300B | 1000G |
| NTv2 50M | 50M | Encoder | k-mer | 4107 | 2048 | 300B | Multispecies |
| GENA-LM | 110M | Encoder | BPE | 32000 | 512 | 1T | HRG+1000G |
| DNABERTv2 | 117M | Encoder | BPE | 4096 | 128 | 262B | Multispecies |
| Caduceus | 8M | Encoder | Char | 12 | 131K | 35B | HRG |
| Mistral | 580M | Decoder | Char | 12 | 4096 | 150B | 1000G |

Table 1: **Description of models evaluated in this study.** The analyzed models differ in architecture, pretraining objective, tokenizer, model size, and pretraining dataset. We analyze the pretrained models and their randomly initialized counterparts. *#Tokens* refers to the number of tokens seen by the model during the pretraining. *Data* refers to the pretraining dataset source.

the tokenizer from characters to 6-mers increases average downstream MCC by $\approx 0.19$ despite a slightly worse pretraining loss.

Finally, we assessed the models on one of the most practically important applications for genomics: detecting subtle genomic variations (Section 3.5). This scenario requires models to be highly sensitive to single nucleotide changes within long sequences. In our evaluation, we find that most pretrained GFMs show limited sensitivity to these variants. For instance, even when up to half of the nucleotides in a DNA sequence are changed, some GFMs still produce embeddings with cosine similarity above 0.99, and ClinVar log-likelihood ratio tests yield AUROCs close to 0.5 (Table 4 and Table 5). These results suggest that, in their current form, the evaluated GFMs provide limited utility for applications that rely extensively on subtle mutation signals, including variant pathogenicity prediction, eQTL (Zhou & Troyanskaya, 2015), sQTL (Garrido-Martín et al., 2021), and phenotype prediction (Lello et al., 2018).

Our results call for a more careful view of current unsupervised pretraining methods in genomics, suggesting that simply adapting NLP techniques is not yet sufficient to obtain broadly useful 'foundation' models on regulatory classification tasks. Rather than continuing to invest substantial computational resources in existing pretraining methods, we advocate for critically rethinking the fundamental building blocks of genomic foundation models. This includes developing biologically-informed tokenization strategies and establishing new robust benchmarks that comprehensively test for the understanding of genomic mechanisms.

## 2 MODELS

We selected six recently published GFMs for evaluation and also trained our own version of the Mistral (Jiang et al., 2023) model on 50 samples from the 1000 Genomes dataset (Consortium et al., 2015). The models in our analysis exhibit significant diversity in their architectures, pretraining objectives, tokenizers, model sizes, and pretraining datasets. Our model selection includes both encoder and decoder architectures, transformer-based and state-space models, with model sizes ranging from 450K to 580M parameters. Interestingly, our Mistral outperforms all other previous GFMs on many tasks. We attribute the success of Mistral to an advanced architecture recipe which includes RoPE, big embedding dimension and character tokenizer. Table 1 lists model configurations; model descriptions are in the Appendix.

We excluded the EVO model (Nguyen et al., 2024) from our analysis as it was trained on bacterial genomes and performed poorly in our preliminary tests on the Nucleotide Transformer Benchmark.

## 3 EXPERIMENTS AND RESULTS

### 3.1 FINETUNING

To verify the usefulness of pretraining, we finetuned both pretrained and randomly initialized (rand. init.) versions of the models on **Nucleotide Transformer Benchmark** (Dalla-Torre et al., 2024), **Genome Understanding Evaluation (GUE)** (Zhou et al., 2024), and **Genomic Benchmarks**

(Grešová et al., 2023) with exactly the same set of hyperparameters. This set of benchmarks together constitutes **52** genomic classification tasks. In total, we conducted nearly **10,000** finetuning experiments, this considers: seven models, both pretrained and random, evaluated across different tasks, folds, and learning rates.

To ensure robustness, we performed a broad hyperparameter search over learning rate, weight decay, batch size, warm-up steps, LoRA (Hu et al., 2022) vs full finetuning, and others. Because performance was most sensitive to the learning rate, the final run consisted of a sweep over six learning-rate values, and we report the best result obtained. We also found that full finetuning consistently outperformed LoRA (see Table 8 in Appendix), suggesting that it provides the best opportunity for the model to reach the full score.

For each model, we now compare its pretrained checkpoint directly against an identically configured version with random initialization. In Fig. 1, the x-axis shows the average performance of the randomly initialized model and the y-axis shows the corresponding pretrained model on (a) NT histone and enhancer tasks, (b) GUE task categories, and (c) Genomic Benchmarks. Points on the diagonal correspond to identical performance, while the vertical distance from the diagonal measures the gain or loss from pretraining.

Reading Fig. 1 along the x-axis reveals that the strength of the random baseline depends strongly on the tokenizer. For instance, randomly-initialized character-token models (Caduceus, HyenaDNA) are often indifferent or superior than their pretrained counterparts. We also note a near-consistent trend of pretraining helping models trained using BPE or k-mer tokenizer, with the extent of gain ranging from 0.05-0.25 MCC on NT and GUE, with model architecture a strong determining factor of performance. Detailed per-task deltas comparing each pretrained model to the best random baseline are provided in Appendix in Fig. 7. Additionally we also provide separated apples-to-apples comparisons for each model between pretrained and rand. init. versions in Fig. 8 (NT Benchmark), Fig. 9 (GUE), Fig. 10 (Genomics Benchmark) in Appendix.

Even after accounting for pretraining gains, small randomly initialized models remain competitive in absolute terms. On NT Benchmark, for example, a randomly initialized Caduceus (8M parameters) achieves an average MCC of about 0.62 on the most challenging histone and enhancer tasks (Fig. 1A), outperforming several larger pretrained models such as NT-500M, NTv2-50M, and GENA-LM (110M parameters), and often matching or exceeding its own pretrained version. DNABERTv2 (117M parameters) also provides a strong random baseline and, on several histone tasks, outperforms pretrained NT-500M by sizable margins (Table 18).

The results on GUE in the middle part of Fig. 1 demonstrate that randomly initialized Caduceus shows remarkable performance outperforming its pretrained version by 0.114 MCC. For instance, on TF Prediction (Mouse) task in Fig. 9, randomly initialized Caduceus outperforms all 7 pretrained models. Similar trend is observed on Core Promoter Detection Group where rand. init. Mistral outperforms all pretrained models including its own pretrained version. In general on GUE benchmark the best randomly initialized model outperforms five to seven pretrained models (Fig. 7). In Genomic Benchmarks similar trend of competitiveness of rand. init. models can be observed (Fig. 10).

The results across all three benchmarks demonstrate that while pretraining improves some GFMs relative to their own random initialization, we also identified several randomly initialized models like Caduceus, DNABERTv2, and HyenaDNA that can match or exceed pretrained performance across a wide range of tasks.

Unlike foundation models in other domains, such as computer vision (Radford et al., 2021) and NLP (Brown et al., 2020), where pretraining typically leads to significant improvements in downstream task performance, the current pretraining strategies in genomics are barely able to outperform randomly initialized models. Moreover, even in cases where pretrained models maintain an advantage, the gains from pretraining are surprisingly small - typically within 2–3%. Together, these modest gains may not justify the large amounts of compute required for pretraining in genomics (Dalla-Torre et al., 2024), especially for these commonly used fine-tuning tasks.

## 3.2 FINETUNING WITH LIMITED DATA

To examine whether pretraining is more beneficial when labels are scarce, we repeated finetuning for four representative models: NT-50M (k-mer), DNABERT-2 (BPE) and GENA-LM (BPE), and

| Tokenizer | Pretrain | Char tokenizer by default | | | Subword tokenizer by default | | | |
| --- | --- | --- | --- | --- | --- | --- | --- | --- |
| | | Mistral | HyenaDNA | Caduceus | NTv2 50M | GENA-LM | DNABERTv2 | NT 500M |
| default | ✓ | 0.730 | 0.638 | 0.423 | 0.679 | 0.704 | 0.654 | 0.662 |
| default | ✗ | 0.667 | 0.690 | 0.674 | 0.482 | 0.574 | 0.651 | 0.603 |
| char | ✗ | 0.666 | 0.690 | 0.674 | 0.642 | 0.668 | 0.696 | 0.669 |
| +larger embed dim | ✗ | 0.700 | 0.753 | 0.717 | 0.703 | 0.684 | 0.708 | 0.678 |
| pretrained − random | | **3.0%** | **-11.5%** | **-29.4%** | **-2.4%** | **2.0%** | **-5.4%** | **-1.6%** |

Table 2: **Biotype classification** F1 scores for pretrained vs. randomly initialized models. Optimizing the tokenizer and embedding dimension (rows 3-4) allows most random models to surpass their pretrained counterparts (last row).

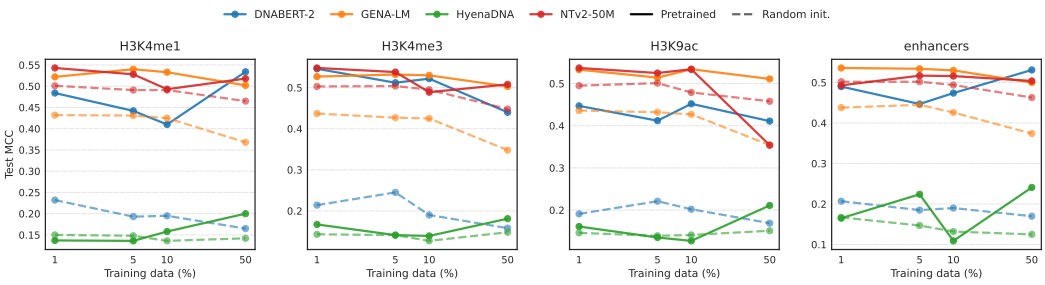

Figure 3: **Pretraining benefit in low-resource regimes depends on tokenization.** Performance curves for H3K4me1 and Enhancers at 1%, 5%, 10%, and 50% data fractions. Subword models (DNABERT-2, GENA-LM) show consistent gains from pretraining (solid / pretrained vs dashed / random lines), while the char-level model (HyenaDNA) often shows smaller benefits.

HyenaDNA (character), on four NT Benchmark tasks (H3K4me1, H3K4me3, H3K9ac, enhancers) using 1%, 5%, 10%, and 50% of the training labels. For each combination of model, dataset, and data fraction, we performed a full hyperparameter sweep over four learning rates (1e-5, 5e-5, 1e-4, 5e-4) trained for 30 epochs, ensuring that both pretrained and random baselines were compared at their optimal settings. The results are displayed in Fig. 3. Pretraining yields consistent and sometimes large gains for the BPE models across all label fractions (with the largest absolute improvements at 1–5%), whereas the character-level HyenaDNA model derives much smaller and sometimes negligible gains from pretraining in the same regime. This confirms that the tokenizer-dependent pattern persists, and is more pronounced, in low-resource settings.

### 3.3 TOKENIZER EFFECT ABLATION

To causally isolate the effect of tokenizer on model performance, we performed a controlled ablation study. We pretrained two identical HyenaDNA models (same model size, same Human Reference Genome dataset, same # of training iterations) differing *only* in their tokenizer; one using a character tokenizer and the other using a 6-mer tokenizer. Both models were then finetuned on a representative subset of the NT Benchmark, specifically the enhancers, H3K4me3, and H3K9ac tasks, and we report the performance averaged across these three tasks. Results are presented in Table 3.

The results in Table 3 offer a critical insight. Although the character-level model achieved a *lower* pretraining loss (1.180 vs 1.215), the k-mer model significantly outperformed it on downstream tasks (+0.187 MCC advantage). This finding demonstrates that pretraining perplexity can be a misleading proxy for downstream task performance. The k-mer tokenization may be providing a useful inductive bias by forcing the model to learn representations of motif-like sequences from the outset, a process that is likely translating to more effective performance on these functional genomics tasks.

### 3.4 FEATURE EXTRACTION

The biotype classification task assesses the quality of features extracted from both pretrained and rand. init. models. Unlike in finetuning, this task does not involve updating model weights. In other

| Metric | Character | K-mer | Difference |
|---|---|---|---|
| Pretraining Loss ↓ | **1.180** | 1.215 | Character better |
| H3K4me3 (MCC) ↑ | 0.138 | **0.323** | K-mer +0.185 |
| H3K9ac (MCC) ↑ | 0.141 | **0.349** | K-mer +0.208 |
| Enhancers (MCC) ↑ | 0.139 | **0.305** | K-mer +0.166 |
| Avg. Downstream MCC ↑ | 0.139 | **0.326** | **K-mer +0.187** |

Table 3: **Tokenizer Ablation.** Comparison of two identical HyenaDNA models pretrained with different tokenizers. Despite the character model achieving lower pretraining loss, the k-mer model vastly outperforms it on all downstream tasks.

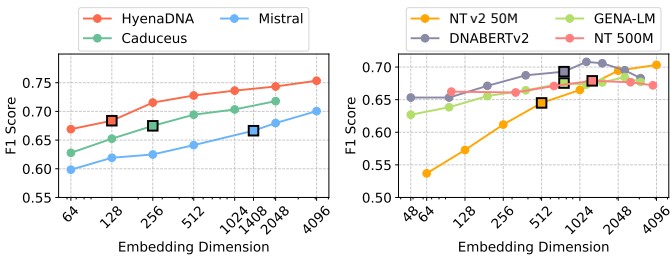

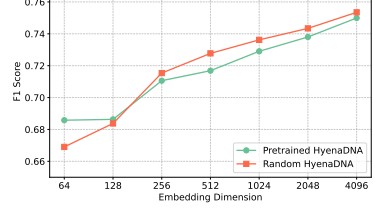

Figure 4: Performance of rand. init models with char tokenizer improves as the embed_dim increases. Squares mark default dimensions.

Figure 5: F1 for pretrained/ random init HyenaDNA on the biotype task; except at d=64, random >pretrained over embed. dims.

words, *embeddings for randomly initialized models were extracted without any finetuning and were entirely based on their initial random weights.*

Using sequences and biotype labels from the Gencode (Harrow et al., 2012), we extracted features from models with frozen weights and applied max pooling along the token dimension. These pooled features were then used to train an XGBoost classifier to predict among nine biotype labels. See Appendix for dataset details.

We observed that the choice of tokenizer significantly impacts the performance of rand. init. encoder models. In particular, switching these models from their default k-mer or BPE tokenizers with large vocabularies (Table 1) to a char tokenizer that has only four tokens substantially improved their performance (3rd row of Table 2, right part). For example, for NTv2 50M, the performance increased from 0.48 to 0.64. Char tokenizer is standard for decoder models, hence the identical results in the second and third rows of Table 2 for decoder models. This improvement likely stems from random models' difficulty handling large vocabulary sizes. Initially, the random HyenaDNA model achieved the highest F1 score (0.69) among random models despite its small 128-dimension embeddings, prompting us to investigate how embedding size affects performance. We tested various embedding dimensions across all models, keeping other parameters constant and ensuring divisibility by the number of attention heads. Complete embedding configurations are detailed in the Appendix.

To decouple embedding dimension from initialization (random vs pretrained), we pretrained new HyenaDNA models from scratch across a full range of embedding dimensions, and compared it to its randomly initialized counterpart under the identical encoder, tokenizer (character), pooling, and classifier setup. The paired curves in Fig. 5 show that pretraining provides an advantage only at very small width (d=64); by d=128 the gap closes, and for all larger widths the randomly initialized model matches or exceeds the pretrained model. Thus, on biotype, the effect of pretraining is brittle with respect to model capacity.

In a related experiment, we find that (Fig. 4) as the embedding dimension of randomly initialized models increases, their performance improves, for nearly all of the models we tested. HyenaDNA shows consistent improvements, reaching an F1 score of nearly 0.75 at 4096 dimensions. NTv2 50M exhibited a more dramatic improvement, with its F1 score rising from 0.53 to 0.71. Further, on GUE histone tasks (Appendix – Table 13), random HyenaDNA (d=2048) is best on 9/10 tasks, again indicating that capacity interacts strongly with initialization. Also, as shown in the fourth row of Table 2, increasing the embed. dim and using a char tokenizer allowed randomly init. models to outperform pretrained in 5 out of 7 cases.

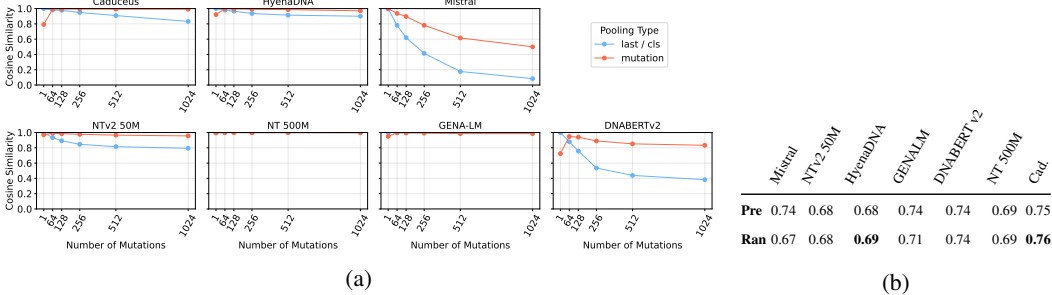

Figure 6: (a) *Mutation sensitivity*: cosine similarity between reference and mutated sequences remains high under both pooling schemes, indicating limited sensitivity to SNPs. (b) *Ancestry classification (F1)*: random Caduceus (0.76) is highest; random HyenaDNA (0.69) slightly exceeds its pretrained counterpart; others are comparable.

## 3.5 GENOMIC VARIATION

This section transitions from functional element classification to genomic variation tasks, analyzing mutations like single nucleotide polymorphisms (SNPs), insertions, and deletions between individuals. While functional elements remain largely consistent across populations, genomic variations are individual-specific and can significantly impact phenotype and disease risk. These tasks present a unique challenge for GFMs, which must detect and interpret subtle sequence differences, often down to single nucleotide changes, to understand human genetic diversity and its health implications.

**Ancestry prediction** is a multilabel classification task that predicts an individual's ancestry using a small portion of their genome. We constructed an ancestry dataset from 1000G data (Consortium et al., 2015), using HG38 and applying mutations from each 1000G sample to obtain 32K-base consensus sequences. These sequences differ by  0.5% of positions, with an average of 33 variants (SNPs, insertions, and deletions). Embeddings generated from these sequences were used as features for XGBoost classification.

When generating the dataset, we selected eleven different regions of the genome, treating each as a separate fold, and evaluated our models on each region independently, reporting average (Fig. 6 (b)). See Appendix for dataset details.

Results in Fig. 6 (b) show that randomly initialized models generally match pretrained models' performance. Only Mistral and GENA-LM showed marginal improvement with pretraining (F1 difference: 0.07 and 0.03). Caduceus achieved the highest F1 score (0.76) in both random and pretrained versions. The NT 500M model, despite being trained on 1000G variants, showed no advantage over its random initialization. This performance pattern could stem from two factors: the high masking probability (15%) in masked language modeling, which exceeds the natural mutation rate (0.5%), and the k-mer tokenization (6 nucleotides) that poorly captures single nucleotide variations.

**Mutation Sensitivity Analysis.** We further investigated models' limited ancestry prediction performance and assess their capability in capturing subtle genomic variations, at the SNP-level. These experiments verify the models' ability to detect differences between reference sequences and sequences with inserted SNPs. We focused exclusively on SNPs to eliminate sequence length as a confounding factor.

We measure the cosine similarity between the embeddings of the reference DNA sequence and the embeddings of the same sequence with SNPs introduced, using both global last / cls pooling and mutation-site pooling that aggregates representations only at tokens overlapping the SNPs. Lower similarity scores indicate better model sensitivity in detecting biologically significant changes, while high scores suggest the model fails to distinguish important genetic variations.

To ensure robustness, we conducted this experiment by sampling twenty-five 1024-length sequences from chromosomes 7, 11, 12, 17, and 19 in HG38, with five sequences per chromosome. The 1024-length was chosen to avoid chunking effects, ensuring it fits within all models' context windows. For each sequence, we created variants by introducing mutations at increasing frequencies (1, 64, 128, 256, 512, 1024) at random positions. Embeddings for both the reference and mutated sequences

| Gene | NT 500M | NTv2 50M | HyenaDNA | Mistral | Caduceus | GENA-LM | DNABERTv2 |
|------|---------|----------|----------|---------|----------|---------|-----------|
| **BRCA2** | 0.511 | 0.478 | 0.439 | 0.495 | 0.505 | 0.408 | 0.535 |
| **CFTR** | 0.442 | 0.365 | 0.454 | 0.345 | 0.442 | 0.421 | 0.536 |

Table 4: Near-random AUROC scores from log-likelihood ratio tests on ClinVar variants, indicating poor model sensitivity to clinically relevant mutations.

| Gene | Mutation Type | NT 500M | NTv2 50M | DNABERTv2 | HyenaDNA | Mistral | GENA-LM | Caduceus |
|------|---------------|---------|----------|-----------|----------|---------|---------|----------|
| **TP53** | **Benign** | 0.985 | 0.991 | 0.995 | 0.999 | 0.976 | 1.000 | 0.985 |
| | **Pathogenic** | 0.983 | 0.993 | 0.996 | 0.999 | 0.988 | 1.000 | 0.990 |
| **BRCA2** | **Benign** | 0.999 | 0.984 | 0.964 | 0.996 | 0.907 | 0.996 | 0.996 |
| | **Pathogenic** | 1.000 | 0.984 | 0.955 | 0.999 | 0.981 | 1.000 | 0.973 |
| **CFTR** | **Benign** | 1.000 | 0.998 | 0.998 | 1.000 | 0.999 | 1.000 | 0.999 |
| | **Pathogenic** | 1.000 | 0.999 | 0.998 | 1.000 | 0.996 | 1.000 | 0.999 |

Table 5: **Gene-specific Variant Detection.** Distinguishing benign vs. pathogenic variants in TP53, BRCA2 and CFTR genes. Lower values are better.

are then generated using different methods: last / cls tokens (based on the decoder or encoder), and pooling the tokens only at mutation sites (denoted as "mutation" in Fig. 6). These two approaches were tried to cover different ways of interpreting the embeddings generated by the model during downstream evaluations.

Fig. 6 illustrates the cosine similarity between reference and altered sequences across different pooling types. Despite the models using different tokenizers, the results are generally poor. For most models, both last / cls and mutation pooling produced high cosine similarity values even for a single mutation regime, typically 0.9 or higher. As the number of mutations increases, the cosine similarity also tends to increase, presumably due to averaging effect. Among the models tested, the Mistral-based DNA model showed the lowest cosine similarity for last pooling and relatively low cosine similarity for mutation pooling. In contrast, GENA-LM and NT 500M produced high cosine similarity scores close to 0.999 for both pooling types. These results indicate that most models are not significantly affected by mutations, thereby highlighting their limited ability to detect subtle sequence alterations, irrespective of their tokenizers.

**ClinVar Experiments.** To further investigate the sensitivity of GFMs to sequence alterations, we conducted additional experiments using ClinVar data (Landrum et al., 2014), which includes genetic variations among individuals. These experiments aim to verify our previous findings in a more realistic setting, using real-world genetic variations from ClinVar. We analyze the TP53, BRCA2 and CFTR genes and obtained their gene sequences from NCBI (Sayers et al., 2022).

First, we filtered the variants to include only exonic mutations. This ensures a focus on mutations that affect protein-coding regions, which are of greatest interest in clinical genetics. Next, we categorized the variants into two groups based on clinical significance: benign and pathogenic. The benign group included variants labeled as 'Benign', 'Likely benign', or 'Benign/Likely benign', while the pathogenic group comprised variants classified as 'Pathogenic', 'Likely pathogenic', or 'Pathogenic/Likely pathogenic'. This grouping enables us to compare the model's sensitivity to mutations with different clinical impacts.

After preprocessing the data, we take five chunks of 1024 base pairs for each gene independently that have both benign and pathogenic mutations. We created three sequences for each chunk: the original reference, one with only pathogenic mutations, and one with only benign mutations. The distribution of mutations is shown in the Appendix.

This variation in mutation density allows us to observe the model's sensitivity across different levels of sequence alteration. For each chunk, we applied max pooling to the model outputs and computed the cosine similarity between the reference sequence and both the benign and pathogenic versions. Finally, we averaged cosine similarity over five selected chunks. The results presented in Table 5 showed consistently high similarity scores across all models and mutation types, indicating the consistent failure of models to reflect genomic variance in their embeddings.

**Log-Likelihood Ratio Analysis.** To further probe the sensitivity of the models to single-nucleotide changes, we evaluated them using log-likelihood ratios, following the methodology of Benegas et al. (2025a). This approach allows us to assess each model's ability to distinguish between alleles without any finetuning. Specifically, for a single-nucleotide variant defined by a reference base (REF) and an alternative base (ALT) at a given position, we compute the log-likelihood ratio: $\log \frac{P(\mathrm{ALT})}{P(\mathrm{REF})}$. For encoder-only models, we follow (Benegas et al., 2025a) and approximate these log probabilities using the softmax scores at the masked variant position, holding the surrounding context fixed. We applied this method to each pathogenic variant in the BRCA2 and CFTR genes from our ClinVar dataset. Log probabilities for the mutated and reference nucleotides were computed directly from the pretrained models at the specific mutation sites. We then used these log-likelihood ratio values to compute AUROC scores for distinguishing pathogenic from benign variants.

The results, presented in Table 4, show that model performance remained near random chance, with AUROC scores between 0.345 and 0.536. This finding further supports our earlier conclusions, providing site-specific evidence that existing pretrained GFMs have limited sensitivity to clinically significant mutations.

## 4 DISCUSSION

We evaluated seven Genomic Foundation Models (GFMs) across 52 tasks spanning three standard benchmarks under finetuning, frozen–feature, and variant sensitivity settings, and covering regulatory, functional, and structural genomics tasks. Two consistent observations emerge. First, the *random baseline* is strongly influenced by tokenization, and the character-tokenized models often attain high performance without pretraining, matching or surpassing much larger pretrained subword models (Sec. 3.1). Second, the *benefit of pretraining* is conditional; we do observe it for subword tokenizers and model/recipe combinations (that pose a harder from-scratch representation learning problem), but find the pretraining gains are inconsistent for character tokenizers. The paired comparison in Fig. 1 makes these patterns explicit by plotting each model's pretrained score against its own randomly initialized counterpart.

**What drives the baseline, and when does pretraining help?** Reading Fig. 1 left-to-right (random) and bottom-to-top (pretrained) shows that character tokenization provides a strong starting point. Subword models (k-mer/BPE) frequently gain more from pretraining, whereas character models display smaller or variable gains. Architecture matters as well – the character-tokenized Mistral benefits from pretraining, indicating that tokenizer, architecture, and scale jointly determine whether pretraining adds signal rather than noise. This tokenizer dependence persists in label-scarce regimes (Fig. 3). With only 1–5% of labels, subword models (e.g., DNABERT-2, GENA-LM) show improvements from pretraining (e.g., DNABERT-2 gains ∼+0.25 MCC on H3K4me1 at 1%), while the character model (HyenaDNA) often shows negligible or negative transfer. In short, pretraining helps most when it amortizes a difficult token-representation problem that supervised finetuning alone cannot solve within budget.

**Mechanism – tokenizer inductive bias and a causal test.** To ascertain the role of tokenizers in leading to the results we observed, we further fixed the architecture and training recipe and changed only the tokenizer for results reported in Table 3. Despite achieving a *lower* pretraining loss, the character-tokenized model yielded substantially *worse* downstream MCC than its k-mer counterpart (+0.187 average advantage for k-mer). Two implications follow. (i) Pretraining loss can be a poor proxy for discriminative genomics: language-model objectives capture token predictability that does not necessarily align with downstream labels. (ii) Tokenizer inductive bias can dominate downstream performance independently of loss. Intuitively, these can be explained as follows. A compact character vocabulary creates an easy input space for randomly init. models, producing a strong baseline; whereas large subword vocabularies create a sparse, hard input space where pretraining confers more value by learning token representations that finetuning alone struggles to form.

**Feature quality at zero training and evaluation hygiene.** In the frozen-feature setting one might expect pretrained features to dominate, since the random arm receives no weight updates. Instead, modest architectural choices (character tokens and increased embedding dimension) make random features competitive and often superior (Table 2; per-task details in the Appendix). Further, under a matched HyenaDNA architecture and a full embedding dimension sweep, pretraining helps only at d=64 and loses its advantage from d=128 upward (Fig. 5), indicating that capacity rather

than pretraining drives most of the gains on that task. Together with our LoRA vs. full-finetuning comparison (Table 8), we arrive at two main observations: (i) well-tuned random baselines (hyperparameters tuned, with lightweight architectural choices) can achieve competitive representation quality; and (ii) cross-modal comparisons where random baselines are tuned with LoRA are likely under-reporting performance for random baselines.

**Variant sensitivity is a central gap.** Many practical applications hinge on single-nucleotide differences. Across pooling schemes, including mutation-site pooling, embedding similarities between reference and mutated sequences remain high (Fig. 6), and ClinVar log-likelihood ratio tests yield AUROC values near chance (Table 4). Character-tokenized models (e.g., Mistral, Caduceus) sometimes show lower similarity or stronger ancestry classification, but effect sizes are small. These findings indicate that, regardless of pretraining status, common objectives and tokenizers do not reliably encode allele-level information needed for pathogenicity, eQTL/sQTL, or related tasks. This points to the need for mutation-aware objectives, tokenization that preserves single-base signals, and evaluation tasks that directly stress allele-level sensitivity.

**Limitations and Outlook.** Our study targets discriminative sequence classification and frozen-feature quality. Several evaluated models also have short context windows (128-1024), which limits our ability to conduct long-range experiments often requiring 100k or longer context. We do not test generative sequence design or long-range modeling tasks such as gene-expression regression or enhancer–promoter linkage, where specialized supervised models and new generative architectures (Evo2 (Brixi et al., 2025), GenomeOcean (Wu et al., 2025), Enformer-style models) remain strong baselines.

*Limitations*. (i) We analyze seven GFMs; broader coverage (e.g., quantized or graph-based models) is outside scope. (ii) Variant sensitivity is evaluated with cosine similarity and site-wise LLR on selected genes; further diagnostics (e.g., per-base in-silico mutagenesis with attribution maps; comparisons to CADD/Enformer; eQTL/sQTL ground truths) are natural extensions. (iii) Our mechanism study isolates tokenizer effects; other factors (masking schedules, positional encodings, curriculum) could be grounds of evaluation for future work. (iv) We focus on maximizing performance of each model, and therefore limit our analysis to default precision. Extension to quantized models (e.g., GERM (Luo et al., 2025)) could be considered by the future studies.

*Outlook*. Three practical paths follow from these results. **(i) Short-range classification** – compact, character-tokenized models are strong and compute-efficient random baselines; reporting well-tuned random baselines should become standard practice. **(ii) Subword models and label-scarce regimes** – expect gains from pretraining, but align the objective with use (e.g., mutation-aware masking, contrastive signals anchored at variant loci, and multi-scale local-global supervision). **(iii) Variant-centric applications** – clear evidence that the community should prioritize tokenizer and objective function redesign before further scaling. In parallel, community benchmarks that couple allele-level probes with long-range regulatory tasks, and require transparent architectural controls, will better track genuine representation gains rather than proxy loss reductions.

## 5 CONCLUSION

Our comprehensive evaluation of GFMs highlights significant limitations in current pretraining approaches. Across standard classification benchmarks, pretrained GFMs offer at most modest, tokenizer- and architecture-dependent advantages over tuned randomly initialized counterparts. We also found that existing GFMs exhibit insufficient sensitivity to variants, which limits their utility in tasks requiring variant interpretation. We identify areas for methodological refinement, including optimizing masking approach, employing character-level tokenization, and designing specialized architectures better attuned to biological sequence complexity. Additionally, our extensive hyperparameter optimization helped ensure the robustness of these conclusions. While genomic pretraining still holds promise, particularly in specialized generative contexts, realizing the vision of broadly applicable, clinically relevant GFMs will require a fundamental reassessment of current practices. We hope our findings encourage the genomic modeling community to develop more biologically informed approaches, rigorous benchmarks, and targeted strategies, ultimately bridging the gap between computational advancements and tangible biomedical impacts.

## REPRODUCIBILITY STATEMENT

Our code is available at github.com/m42-health/gfm-random-eval. In our study we used such publicly available datasets as:

- 1000 Genomes Project VCF files can be accessed at International Genome Sample Resource (IGSR) (Fairley et al., 2020).
- GRCh38 reference genome assembly can be downloaded from (National Center for Biotechnology Information (NCBI), 2024).
- Gencode (Frankish et al., 2019) gene annotation used for biotype labeling.
- NT Benchmark datasets are introduced in (Dalla-Torre et al., 2024).
- Genome Understanding Evaluation (GUE) multi-species benchmark is introduced in (Zhou et al., 2024).
- Genomic Benchmarks is introduced in (Grešová et al., 2023).
- ClinVar variant records for TP53, BRCA2 and CFTR and corresponding gene sequences are downloaded from NCBI (National Center for Biotechnology Information (NCBI), 2024).

Details on training hyperparameters for evaluations are provided in the Appendix. Finally, the Appendix includes a description of the computational environment used for our experiments.

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

# A  APPENDIX

## A.1  RELATED WORKS

**Genomic Foundation Models.** Encoder-only approaches have proven effective in sequence prediction tasks, using k-mer tokenization (Ji et al., 2021; Dalla-Torre et al., 2024), Byte Pair Encoding (Zhou et al., 2024; Sanabria et al., 2024), and learned tokenization strategies (Qiao et al., 2024; Li et al., 2024) to enhance efficiency and manage longer sequences. Certain encoder architectures

have been enhanced with recurrent memory mechanisms (Fishman et al., 2025) to capture long-range dependencies more effectively, while others utilize whole-genome alignments (Benegas et al., 2025a) to incorporate evolutionary context. More recent work has explored pan-genome graph representations (Zhang et al., 2024) to better capture genetic variation diversity.

Meanwhile, decoder-only architectures have shown potential by integrating structured state-space models (Nguyen et al., 2023; Schiff et al., 2024), achieving competitive performance with minimal parameters and supporting long context lengths. Hybrid architectures (Nguyen et al., 2024; Brixi et al., 2025), incorporating both attention and state-space blocks, have emerged, demonstrating great generative capabilities spanning from molecular to genome scales. Our work introduces a GFM based on Mistral architecture (Jiang et al., 2023) and performs performance analysis of the most recent GFMs.

**Genomic Foundation Models Analysis.** It was shown that k-mer embeddings pretrained on random DNA sequences can reach similar performance to those trained on the real-world biological data (Zhang et al., 2023). Another study found that character tokenization outperforms other methods in state-space models (Lindsey et al., 2025). Evaluation of GFMs across the BEND benchmark reveals that they capture limited information on long-range features (Marin et al., 2024). It was also shown that mean pooling improves performance of GFMs for genomic sequence classifications and closes the performance gap between them (Feng et al., 2025). Pretrained DNA models were benchmarked (Tang et al., 2024) showing they do not offer great advantage over conventional machine learning methods. In contrast to this study, our analysis includes finetuning and variant-based tasks, more models and also shows that randomly initialized models can be better as feature extractors.

## A.2 MODELS

We use the following GFMs in our analysis:

- **HyenaDNA** (Nguyen et al., 2023): Decoder-only state-space model with 450K parameters. Uses character tokenizer and was pretrained on the Human Reference Genome with a 1024 base pair sequence length.
- **Caduceus** (Schiff et al., 2024): Decoder-only model with 8M parameters. Trained on sequences of 131k base pairs on HRG. Combines a bidirectionally equivariant decoder with character tokenizer.
- **Mistral** (our version): Decoder-only transformer model with 580M parameters. Uses character tokenization and was trained on the 1000 Genomes dataset (Consortium et al., 2015).
- **Nucleotide Transformer** (Dalla-Torre et al., 2024): Encoder-only model presented in two versions: a 500M parameter model trained on the 1000 Genomes Project data and its v2 with 50M parameter model trained on multispecies data. Both use k-mer tokenization.
- **GENA-LM** (Fishman et al., 2025): Encoder-only model with 110M parameters. Employs BPE tokenizer and was pretrained on the HRG with 1000G augmentations.
- **DNABERTv2** (Zhou et al., 2024): Encoder-only model with 117M parameters. Uses BPE tokenization and was trained on multispecies data.

## A.3 RANDOM WEIGHT INITIALIZATION

We initialized the model weights following a procedure using standard Hugging Face Transformers library (Wolf et al., 2020) initialization methods:

- **For Linear Layers:** Weights were initialized from a normal distribution $\mathcal{N}(0, 0.02)$, biases were initialized to zero.
- **For LayerNorm:** The scaling factor (gamma) was initialized to 1. The bias term (beta) was initialized to 0.
- **For Embedding Layers:** Embeddings were initialized from the same normal distribution $\mathcal{N}(0, 0.02)$.

For Caduceus and HyenaDNA we performed prenorm residual rescaling, which is the default weight initialization procedure for these models. Biases for linear layers were initialized as zeros.

### A.4 MISTRAL PRETRAINING

We pretrain a Mistral model on 50 random individual samples from the 1000 Genomes Project. Table 6 provides the Mistral configuration details and Table 7 provides the Mistral training configuration. Specifically, reverse complement of sequences formed with Genome1000 VCFs is used with a probability of 0.5. All the chromosomes (chr1 - chrX) are used for sequence formation from 50 individuals. Individuals are sampled in a stratified way, 10 from each superpopulation. We filtered out sequences where number of unknown nucleotides was more than half of sequence length. Total number of tokens is 150B. We used PyTorch FSDP framework. Training was done on GPU cluster using 6 nodes with 8 Nvidia H100 GPUs per node.

| config | value |
| --- | --- |
| num_hidden_layers | 16 |
| num_attention_heads | 16 |
| hidden_size | 1408 |
| vocab_size | 12 |
| intermediate_size | 7168 |

Table 6: **Mistral model architecture.**

| config | value |
| --- | --- |
| tokenizer | character |
| sequence_len | 4096 |
| num_epochs | 1 |
| initial_lr | 7.2e-4 |
| final_lr | 4.2e-5 |
| optimizer_momentum | $\beta_1, \beta_2 = 0.9, 0.95$ |
| lr schedule | cosine with warmup |
| batch_size | 64 |
| num_genomes | 50 |

Table 7: **Mistral training configuration.**

### A.5 COMPUTING INFRASTRUCTURE

We used the following computing infrastructure to run the experiments:

**Hardware Specifications:**

- **GPU Cluster:** 6 nodes with 8 Nvidia H100 GPUs per node (48 H100 GPUs total)
- **GPU Memory:** 80GB

**Software Specifications:**

- **Framework:** PyTorch with FSDP (Fully Sharded Data Parallel)
- **CUDA Version:** 12.1

### A.6 FINETUNING EXPERIMENTS

We use the following datasets for finetuning, more details about them can be found in the corresponding original papers:

- **NT Benchmark** (Dalla-Torre et al., 2024) consists of the following group of tasks histones, enhancers, promoters and splice sites.
- **Genomic Benchmarks** (Grešová et al., 2023) contains several datasets focused on regulatory element classification tasks across three organisms: human, mouse, and roundworm.
- **Genome Understanding Evaluation (GUE)** (Zhou et al., 2024) is a comprehensive multi-species benchmark containing 28 datasets across 7 genomic analysis tasks including promoter detection, transcription factor prediction, splice site detection, etc. with sequence lengths ranging from 70 to 1000 base pairs.

We finetune random and pretrained initializations of the chosen model using the configuration provided in Table 9. In our preliminary experiments, we found that max pooling performed better than cls / last pooling for randomly initialized models while maintaining performance for pretrained, so we used max pooling consistently across all experiments.

Additionally, we perform an ablation for full finetuning vs LoRA (Hu et al., 2022) finetuning and present the result in Table 8. In all the cases, full finetuning outperforms LoRA, suggesting that our

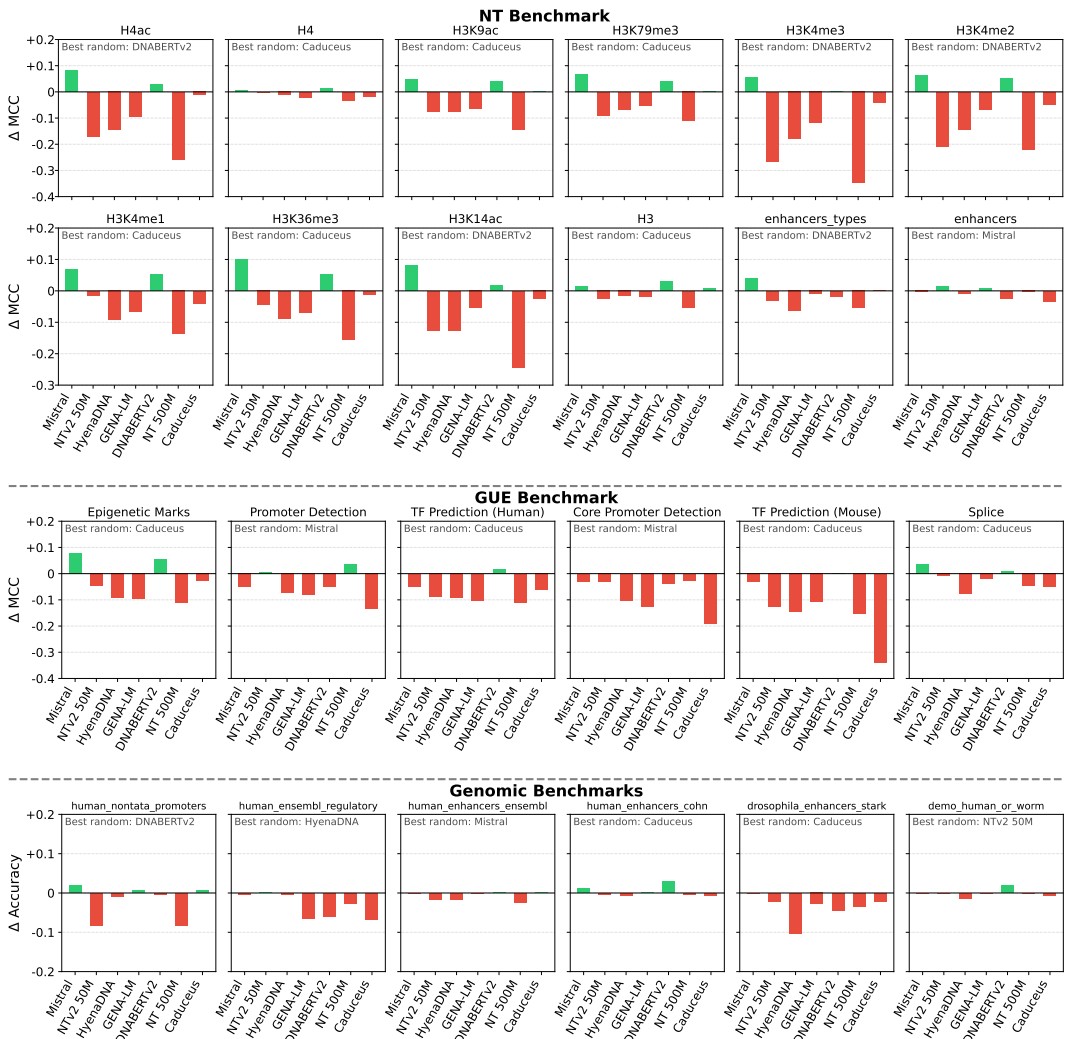

Figure 7: **Pretraining provides limited or no advantage over randomly initialized models across diverse genomic tasks.** We plot the performance difference (Δ MCC or Δ Accuracy) between seven pretrained GFMs and the *best-performing randomly initialized model* for each task. **Red bars** indicate tasks where the randomly initialized model outperformed, whereas **green bars** indicate tasks where the pretrained model performed better. The predominance of red bars provides strong evidence that current pretraining paradigms, adapted from NLP, offer minimal benefit for genomics tasks. Notably, smaller randomly initialized models (e.g., Caduceus) frequently achieve the best performance, challenging the assumption that larger pretrained models are inherently superior.

full finetuning method gives the best chance for the models (both pretrained and random) to achieve their best scores.

| Model | Method | enhancers | H3K4me1 | promoter_all | splice_sites_all |
|---|---|---|---|---|---|
| GENA-LM | LoRA | 0.449 | 0.260 | 0.908 | 0.501 |
| | Full | **0.560** | **0.466** | **0.966** | **0.935** |
| Mistral | LoRA | 0.470 | 0.267 | 0.906 | 0.525 |
| | Full | **0.550** | **0.557** | **0.965** | **0.980** |
| NTv2 50M | LoRA | 0.405 | 0.267 | 0.852 | 0.458 |
| | Full | **0.551** | **0.524** | **0.957** | **0.979** |

Table 8: **LoRA *vs.* full finetuning** on four representative NT tasks. The metric is MCC for enhancers and histone, and F1 for promoters and splice sites. Across all cases full finetuning performs better than LoRA.

| config | value |
|---|---|
| optimizer | AdamW |
| learning_rate | 1e-5, 3e-5, 5e-5, 8e-5, 1e-4, 3e-4 |
| weight_decay | 0 |
| optimizer_momentum | $\beta_1 = 0.9$, $\beta_2 = 0.999$ |
| batch_size | 32 |
| lr schedule | cosine |
| epochs | 20 / 100 |

Table 9: **Hyperparameters for fine-tuning experiments.** For GUE we finetune for 20 epochs for NT Benchmark and Genomic Benchmarks we use 100 epochs.

### A.7 DETAILED FINETUNING PERFORMANCE BY TASK SUBGROUP

To showcase the individual performance of randomly initialized models, we present their results on NT Benchmark alongside pretrained models in Fig. 8. For instance, the "Enhancers" subgroup includes all enhancer-related tasks, while the "Histone" subgroup covers all histone tasks, and so on. In addition, we also show this for GUE and Genomic Benchmarks in Fig. 9 and Fig. 10. We also provide results for all models on NT Benchmark in Table 18.

The results presented in Fig. 8 highlight that randomly initialized models can perform remarkably well across all subgroups of the NT Benchmark. In the "Enhancers" subgroup, all randomly initialized models perform comparably to their pretrained counterparts. In histone tasks, the best random models, DNABERTv2 and Caduceus, reach average MCC scores of 0.62 and 0.63, outperforming pretrained NTv2 50M, HyenaDNA, GENA-LM, and NT 500M. In case of randomly initialized Caduceus it also outperforms its own pretrained version.

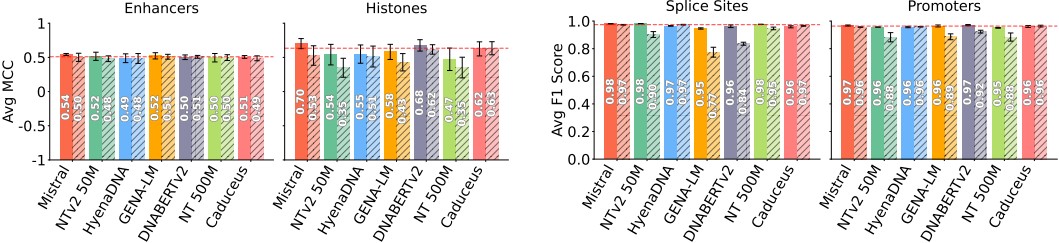

Figure 8: **NT Benchmark performance per subgroup.** Pretrained models (clear bars) vs randomly initialized (dashed bars). Random models are competitive across all subgroups, with random Caduceus outperforming five pretrained models on challenging histone tasks. Red dashed line shows best random model score.

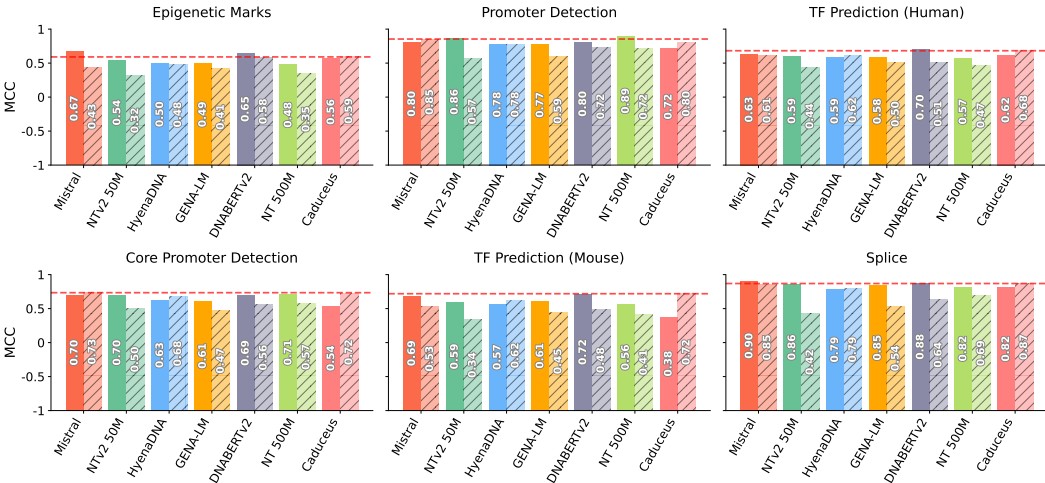

Figure 9: **GUE performance for each model.** Solid bars indicate pretrained model, dashed bars indicate random. Horizontal red dashed line indicates the performance of the best random model. The best rand. init. model is consistently competitive with pretrained models.

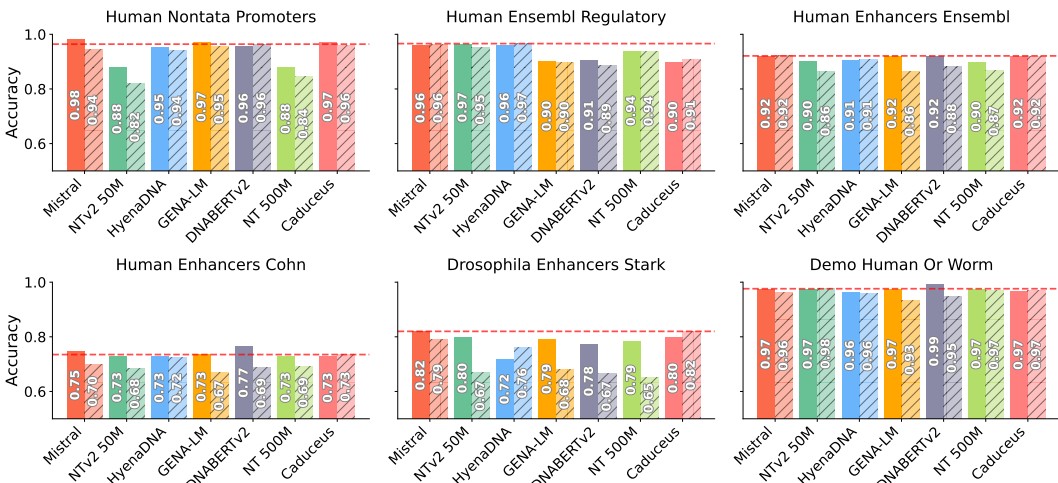

Figure 10: **Genomic Benchmarks performance for each model.** Solid bars indicate pretrained model, dashed bars indicate randomly initialized. Horizontal red dashed line indicates the performance of the best random model.

## A.8 BIOTYPE CLASSIFICATION

Biotype task is a sequence classification task into nine different labels. Our dataset consists of a total of 19605 sequences. The detailed statistics for sequences belonging to each gene type is provided in Table 10. For the supervised training step, we perform a train-test split of 80% : 20% using stratification by class label. We use XGBoost with the hyperparameters provided in Table 12. All metrics are reported on the test set.

In addition, we also perform similar feature extraction experiments on the subset of GUE benchmark displayed in Table 13. Randomly initialized HyenaDNA with large embedding size outperforms all pretrained models.

| Gene Type | Count | Avg Length | Max Length | Min Length |
|---|---|---|---|---|
| TEC | 1056 | 1613.26 | 18662 | 87 |
| lncRNA | 3000 | 32359.59 | 957949 | 87 |
| miRNA | 1879 | 81.89 | 180 | 41 |
| misc RNA | 2212 | 206.49 | 464 | 57 |
| processed pseudogene | 3000 | 798.02 | 12016 | 28 |
| protein coding | 3000 | 69971.51 | 2059620 | 159 |
| snRNA | 1901 | 110.46 | 328 | 50 |
| snoRNA | 943 | 118.86 | 791 | 55 |
| unprocessed pseudogene | 2614 | 5025.27 | 233909 | 28 |

Table 10: **Statistics of biotype genes.**

| Model | Embedding Dimensions |
|---|---|
| HyenaDNA | 64, 128, 256, 512, 1024, 2048, 4096 |
| Caduceus | 64, 128, 256, 512, 1024, 2048 |
| NTv2 50M | 64, 128, 256, 512, 1024, 2048, 4096 |
| DNABERTv2 | 48, 96, 192, 384, 768, 1152, 1536, 2304, 3072 |
| GENA-LM | 48, 96, 192, 384, 768, 1152, 1536, 2304, 3072 |
| NT 500M | 100, 320, 640, 1280, 2560, 3840 |
| Mistral | 64, 128, 256, 512, 1408, 2048, 4096 |

Table 11: **Embedding dimensions for biotype experiments.**

| config | value |
|---|---|
| objective | multi:softmax |
| num_classes | 9 |
| max_depth | 3 |
| learning_rate | 0.1 |
| n_estimators | 1000 |
| eval_metric | mlogloss |
| tree_method | hist |

Table 12: **Biotype XGBoost configuration.**

| Task | HyenaDNA Random ED 2048 | Pretrained | | | | | |
|---|---|---|---|---|---|---|---|
| | | Mistral | HyenaDNA | NTv2 50M | GENA-LM | DNABERTv2 | NT 500M |
| H3 | **0.650** | 0.626 | 0.510 | 0.502 | 0.546 | 0.566 | 0.557 |
| H3K14ac | 0.275 | 0.227 | 0.190 | 0.272 | 0.208 | **0.338** | 0.220 |
| H3K36me3 | **0.408** | 0.267 | 0.252 | 0.330 | 0.321 | 0.397 | 0.308 |
| H3K4me1 | **0.320** | 0.224 | 0.211 | 0.275 | 0.244 | 0.295 | 0.267 |
| H3K4me2 | **0.265** | 0.243 | 0.186 | 0.176 | 0.218 | 0.185 | 0.245 |
| H3K4me3 | **0.207** | 0.126 | 0.105 | 0.147 | 0.113 | 0.189 | 0.121 |
| H3K79me3 | **0.522** | 0.428 | 0.367 | 0.463 | 0.437 | 0.520 | 0.406 |
| H3K9ac | **0.429** | 0.373 | 0.288 | 0.273 | 0.318 | 0.343 | 0.343 |
| H4 | **0.671** | 0.649 | 0.491 | 0.575 | 0.577 | 0.658 | 0.612 |
| H4AC | **0.282** | 0.227 | 0.202 | 0.227 | 0.200 | 0.259 | 0.225 |
| Average | **0.403** | 0.339 | 0.280 | 0.324 | 0.318 | 0.375 | 0.330 |

Table 13: **Feature Extraction on Histone Tasks from GUE.** Embeddings extracted from pretrained and randomly initialized models were used to train an XGBoost classifier. Randomly initialized HyenaDNA with $embed\_dim$ 2048 outperforms every pretrained model on every task except H3K14ac. MCC on test set is reported.

### A.9 ANCESTRY BENCHMARK

Each task is the sequence classification task with five labels, South Asian, European, African, American, East Asian. Each label is a superpopulation from 1000 Genomes dataset. We selected eleven different regions on chromosome with the length of 32K nucleotides, where each region corresponds to a different variant. The start indices with respect to the human reference genome used for sequence construction are provided in Table 14. Each task has 3202 samples.

Training involves two stages: embedding generation from the model of interest and supervised training on the embeddings with XGBoost. During the embedding generation step, sequence embeddings are constructed similarly to the biotype classification task. For the supervised training step, we split the dataset into train, validation and test set with sizes 72%, 8%, and 20% respectively. We use XGBoost with hyperparameters mentioned in Table 15. All metrics are reported on the test set and averaged over eleven tasks across each chromosome.

| chromosome | start position |
|---|---|
| chr1 | 119478211 |
| chr3 | 2015011 |
| chr5 | 85769129 |
| chr7 | 74672986 |
| chr9 | 75197358 |
| chr11 | 62543311 |
| chr13 | 52182164 |
| chr15 | 45995594 |
| chr17 | 36628720 |
| chr19 | 24308808 |
| chr21 | 18354991 |

Table 14: **Sample chromosome positions.**

| config | value |
|---|---|
| objective | multi:softmax |
| num_class | 5 |
| max_depth | 3 |
| learning_rate | 0.1 |
| n_estimators | 1000 |
| colsample_bytree | 0.5 |
| eval_metric | mlogloss |
| tree_method | hist |
| early_stopping rounds | 100 |

Table 15: **Ancestry XGBoost configuration.**

## A.10 CLINVAR EXPERIMENTS

Each chunk used for ClinVar experiments consists of benign and pathogenic mutations. Three types of sequences are formed: reference sequence, sequence with benign mutations, and sequence with pathogenic mutations. The distribution of mutations in these chunks for all three genes is presented in Table 16.

| Chunk Index | TP53 | | BRCA2 | | CFTR | |
|---|---|---|---|---|---|---|
| | Benign | Pathogenic | Benign | Pathogenic | Benign | Pathogenic |
| 1 | 122 | 27 | 138 | 46 | 32 | 18 |
| 2 | 60 | 61 | 268 | 74 | 19 | 11 |
| 3 | 51 | 50 | 187 | 57 | 32 | 30 |
| 4 | 76 | 42 | 35 | 18 | 9 | 13 |
| 5 | 38 | 10 | 37 | 6 | 7 | 11 |

Table 16: **Mutation Data Distribution by Gene and Chunk.** Distribution of benign and pathogenic mutations across different chunks for TP53, BRCA2, and CFTR genes.

## A.11 MODEL CHECKPOINTS

Checkpoints for all the pretrained models were obtained from Hugging Face. Table 17 provides detailed checkpoint IDs which can be loaded using the transformers library.

| Model | Checkpoint |
|---|---|
| NTv2 50M | InstaDeepAI/nucleotide-transformer-v2-50m-multi-species |
| NT 500M | InstaDeepAI/nucleotide-transformer-500m-1000g |
| Caduceus | kuleshov-group/caduceus-ps_seqlen-131k_d_model-256_n_layer-16 |
| HyenaDNA | LongSafari/hyenadna-tiny-1k-seqlen-hf |
| DNABERTv2 | zhihan1996/DNABERT-2-117M |
| GENA-LM | AIRI-Institute/gena-lm-bert-base-t2t |

Table 17: **Checkpoints used for pretrained models.**

| Dataset | Metric | Pretrained | | | | | | | Random | | | | | | |
| --- | --- | --- | --- | --- | --- | --- | --- | --- | --- | --- | --- | --- | --- | --- | --- |
| | | Mistral | NTv2 50M | HyenaDNA | GENA-LM | DNABERTv2 | NT 500M | Caduceus | Mistral | NTv2 50M | HyenaDNA | GENA-LM | DNABERTv2 | NT 500M | Caduceus |
| H4ac | MCC | $0.702 \pm 0.034$ | $0.450 \pm 0.001$ | $0.476 \pm 0.015$ | $0.526 \pm 0.006$ | $0.652 \pm 0.009$ | $0.362 \pm 0.003$ | $0.610 \pm 0.004$ | $0.395 \pm 0.032$ | $0.254 \pm 0.015$ | $0.427 \pm 0.008$ | $0.382 \pm 0.002$ | $0.622 \pm 0.006$ | $0.245 \pm 0.009$ | $0.606 \pm 0.007$ |
| H4 | MCC | $0.802 \pm 0.011$ | $0.792 \pm 0.000$ | $0.783 \pm 0.005$ | $0.773 \pm 0.013$ | $0.809 \pm 0.004$ | $0.762 \pm 0.001$ | $0.778 \pm 0.005$ | $0.760 \pm 0.012$ | $0.593 \pm 0.008$ | $0.787 \pm 0.003$ | $0.619 \pm 0.012$ | $0.669 \pm 0.006$ | $0.629 \pm 0.012$ | $0.795 \pm 0.007$ |
| H3K9ac | MCC | $0.664 \pm 0.004$ | $0.540 \pm 0.009$ | $0.541 \pm 0.004$ | $0.551 \pm 0.008$ | $0.654 \pm 0.004$ | $0.473 \pm 0.005$ | $0.618 \pm 0.005$ | $0.470 \pm 0.005$ | $0.345 \pm 0.007$ | $0.532 \pm 0.012$ | $0.435 \pm 0.018$ | $0.593 \pm 0.002$ | $0.366 \pm 0.016$ | $0.615 \pm 0.020$ |
| H3K79me3 | MCC | $0.753 \pm 0.005$ | $0.596 \pm 0.005$ | $0.616 \pm 0.005$ | $0.633 \pm 0.006$ | $0.725 \pm 0.006$ | $0.574 \pm 0.005$ | $0.688 \pm 0.020$ | $0.666 \pm 0.008$ | $0.435 \pm 0.010$ | $0.576 \pm 0.007$ | $0.505 \pm 0.001$ | $0.670 \pm 0.002$ | $0.445 \pm 0.015$ | $0.685 \pm 0.005$ |
| H3K4me3 | MCC | $0.662 \pm 0.016$ | $0.340 \pm 0.017$ | $0.427 \pm 0.004$ | $0.487 \pm 0.018$ | $0.607 \pm 0.002$ | $0.259 \pm 0.016$ | $0.565 \pm 0.008$ | $0.444 \pm 0.018$ | $0.168 \pm 0.004$ | $0.354 \pm 0.009$ | $0.223 \pm 0.129$ | $0.606 \pm 0.014$ | $0.162 \pm 0.003$ | $0.555 \pm 0.002$ |
| H3K4me2 | MCC | $0.574 \pm 0.024$ | $0.303 \pm 0.004$ | $0.367 \pm 0.012$ | $0.445 \pm 0.018$ | $0.564 \pm 0.005$ | $0.289 \pm 0.012$ | $0.463 \pm 0.007$ | $0.270 \pm 0.001$ | $0.207 \pm 0.009$ | $0.302 \pm 0.006$ | $0.316 \pm 0.010$ | $0.511 \pm 0.000$ | $0.220 \pm 0.004$ | $0.493 \pm 0.002$ |
| H3K4me1 | MCC | $0.603 \pm 0.007$ | $0.518 \pm 0.002$ | $0.441 \pm 0.003$ | $0.468 \pm 0.005$ | $0.585 \pm 0.002$ | $0.397 \pm 0.013$ | $0.493 \pm 0.015$ | $0.442 \pm 0.027$ | $0.260 \pm 0.009$ | $0.421 \pm 0.004$ | $0.334 \pm 0.006$ | $0.499 \pm 0.007$ | $0.254 \pm 0.008$ | $0.534 \pm 0.003$ |
| H3K36me3 | MCC | $0.725 \pm 0.007$ | $0.581 \pm 0.003$ | $0.538 \pm 0.008$ | $0.555 \pm 0.011$ | $0.676 \pm 0.011$ | $0.469 \pm 0.008$ | $0.613 \pm 0.022$ | $0.573 \pm 0.014$ | $0.371 \pm 0.018$ | $0.495 \pm 0.004$ | $0.416 \pm 0.008$ | $0.611 \pm 0.016$ | $0.335 \pm 0.011$ | $0.625 \pm 0.013$ |
| H3K14ac | MCC | $0.724 \pm 0.011$ | $0.516 \pm 0.013$ | $0.516 \pm 0.007$ | $0.587 \pm 0.009$ | $0.659 \pm 0.006$ | $0.396 \pm 0.024$ | $0.615 \pm 0.008$ | $0.578 \pm 0.013$ | $0.281 \pm 0.009$ | $0.462 \pm 0.007$ | $0.403 \pm 0.016$ | $0.641 \pm 0.003$ | $0.259 \pm 0.005$ | $0.637 \pm 0.011$ |
| H3 | MCC | $0.809 \pm 0.003$ | $0.769 \pm 0.006$ | $0.778 \pm 0.007$ | $0.776 \pm 0.005$ | $0.826 \pm 0.012$ | $0.741 \pm 0.013$ | $0.802 \pm 0.012$ | $0.679 \pm 0.012$ | $0.573 \pm 0.008$ | $0.770 \pm 0.005$ | $0.656 \pm 0.001$ | $0.742 \pm 0.009$ | $0.597 \pm 0.008$ | $0.795 \pm 0.002$ |
| enhancers_types | MCC | $0.528 \pm 0.006$ | $0.456 \pm 0.008$ | $0.424 \pm 0.028$ | $0.477 \pm 0.012$ | $0.469 \pm 0.002$ | $0.433 \pm 0.023$ | $0.487 \pm 0.018$ | $0.441 \pm 0.028$ | $0.438 \pm 0.025$ | $0.412 \pm 0.029$ | $0.474 \pm 0.013$ | $0.488 \pm 0.019$ | $0.454 \pm 0.008$ | $0.451 \pm 0.011$ |
| enhancers | MCC | $0.557 \pm 0.003$ | $0.576 \pm 0.010$ | $0.552 \pm 0.009$ | $0.569 \pm 0.010$ | $0.536 \pm 0.009$ | $0.557 \pm 0.027$ | $0.527 \pm 0.012$ | $0.561 \pm 0.004$ | $0.523 \pm 0.016$ | $0.556 \pm 0.022$ | $0.545 \pm 0.015$ | $0.529 \pm 0.002$ | $0.540 \pm 0.011$ | $0.523 \pm 0.024$ |
| splice_sites_all | F1 Score | $0.980 \pm 0.000$ | $0.980 \pm 0.000$ | $0.962 \pm 0.006$ | $0.941 \pm 0.002$ | $0.950 \pm 0.001$ | $0.977 \pm 0.002$ | $0.959 \pm 0.003$ | $0.977 \pm 0.000$ | $0.908 \pm 0.015$ | $0.975 \pm 0.001$ | $0.726 \pm 0.003$ | $0.835 \pm 0.005$ | $0.951 \pm 0.003$ | $0.967 \pm 0.003$ |
| splice_sites_donors | F1 Score | $0.979 \pm 0.001$ | $0.981 \pm 0.002$ | $0.966 \pm 0.003$ | $0.945 \pm 0.003$ | $0.964 \pm 0.002$ | $0.977 \pm 0.001$ | $0.949 \pm 0.009$ | $0.971 \pm 0.001$ | $0.925 \pm 0.006$ | $0.971 \pm 0.001$ | $0.811 \pm 0.008$ | $0.823 \pm 0.003$ | $0.956 \pm 0.003$ | $0.962 \pm 0.004$ |
| splice_sites_acceptors | F1 Score | $0.981 \pm 0.001$ | $0.983 \pm 0.001$ | $0.968 \pm 0.000$ | $0.953 \pm 0.002$ | $0.973 \pm 0.002$ | $0.975 \pm 0.002$ | $0.968 \pm 0.003$ | $0.972 \pm 0.002$ | $0.874 \pm 0.017$ | $0.976 \pm 0.003$ | $0.787 \pm 0.000$ | $0.850 \pm 0.014$ | $0.934 \pm 0.000$ | $0.970 \pm 0.001$ |
| promoter_tata | F1 Score | $0.962 \pm 0.007$ | $0.956 \pm 0.003$ | $0.950 \pm 0.002$ | $0.955 \pm 0.003$ | $0.965 \pm 0.004$ | $0.947 \pm 0.000$ | $0.951 \pm 0.007$ | $0.958 \pm 0.004$ | $0.836 \pm 0.009$ | $0.960 \pm 0.004$ | $0.856 \pm 0.012$ | $0.909 \pm 0.003$ | $0.843 \pm 0.002$ | $0.955 \pm 0.007$ |
| promoter_no_tata | F1 Score | $0.970 \pm 0.003$ | $0.958 \pm 0.002$ | $0.961 \pm 0.000$ | $0.970 \pm 0.002$ | $0.975 \pm 0.000$ | $0.954 \pm 0.001$ | $0.967 \pm 0.002$ | $0.957 \pm 0.001$ | $0.906 \pm 0.002$ | $0.958 \pm 0.001$ | $0.899 \pm 0.003$ | $0.929 \pm 0.001$ | $0.903 \pm 0.002$ | $0.969 \pm 0.001$ |
| promoter_all | F1 Score | $0.971 \pm 0.002$ | $0.957 \pm 0.002$ | $0.960 \pm 0.001$ | $0.969 \pm 0.001$ | $0.973 \pm 0.001$ | $0.953 \pm 0.000$ | $0.965 \pm 0.000$ | $0.955 \pm 0.001$ | $0.907 \pm 0.003$ | $0.958 \pm 0.001$ | $0.903 \pm 0.003$ | $0.932 \pm 0.001$ | $0.908 \pm 0.003$ | $0.968 \pm 0.000$ |

Table 18: **Pretrained and randomly initialized models performance on NT Benchmark.**

