# OpenReview forum: "Tokenization to Transfer: Do Genomic Foundation Models Learn Good Representations?"
_ICLR.cc/2026/Conference — ICLR 2026 Poster_

### Official Review · Reviewer_rYMn · 2025-10-26

**Soundness:** 2
**Presentation:** 3
**Contribution:** 3
**Rating:** 4
**Confidence:** 3

**Summary:**

This paper evaluates the effectiveness of pretraining in Genomic Foundation Models (GFMs). The authors benchmark seven GFMs across 52 genomic tasks, comparing them with randomly initialized counterparts. Results show that randomly initialized models often match or even outperform pretrained GFMs in both fine-tuning and feature extraction settings. The study concludes that current research lacks a clear understanding of how pretraining contributes to model performance and offers new insights into how to achieve more efficient genomic modeling.

**Strengths:**

- Meaningful Perspective: The paper focuses on evaluating the effectiveness of pretraining in GFMs and raises new research questions and perspectives for genomic modeling, providing valuable insights for the community.

- Comprehensive Evaluation: The study benchmarks 7 representative GFMs across 52 diverse genomic tasks, covering structural, functional, and regulatory dimensions.

- Extensive Experimental Setup: The authors conduct detailed comparisons of fine-tuning techniques, including full fine-tuning and LoRA. Also perform extensive hyperparameter searches over learning rate, batch size, and other factors with nearly 10,000 fine-tuning experiments.

**Weaknesses:**

- Potentially Unfair Comparison Setup: Randomly initialized models received embedding dimension optimization, but similar architectural optimization was not performed for pretrained models. What would happen if the pre-trained model also used a character tokenizer and a larger embedding dimension?
- Analysis of Pretraining: The observation that a character-level tokenizer benefits randomly initialized models may indicate design issues in the pretrained models rather than reflecting the true effect of pretraining on genomic models.
- Task Distribution: The paper evaluates 52 tasks, but it is unclear what criteria were used to select them. Are there specific types of tasks like long-range dependency tasks where pretrained models might perform better?
- Generative Tasks: All seven models evaluated in the paper are classification models, and the 52 tasks focus primarily on classification, with no assessment of generative tasks. Although the authors explain their reasons for not using Evo, recent large-scale genomic models such as Evo2[1] and GenomeOcean[2] are designed for generative purposes. Do the conclusions drawn in this study also hold for generative models, or are they only valid when generative models are applied to classification tasks?
- Quantized Model: In this paper, all models are tested under full precision. However, quantized genomic foundation models (GFMs) like GERM [3] also exist. Does the same conclusion hold for these models?

[1] Genome modeling and design across all domains of life with Evo 2.

[2] GenomeOcean: An Efficient Genome Foundation Model Trained on Large-Scale Metagenomic Assemblies.

[3] Fast and Low-Cost Genomic Foundation Models via Outlier Removal.

**Questions:**

See the weaknesses section.

---

> ### Author Response · Authors · 2025-11-27
> **Response to Reviewer – rYMn**
>
> **Please also see the “COMMENT FOR ALL REVIEWERS” for a high‑level summary of changes.**
>
> We sincerely appreciate the reviewer's feedback. We respond to the weaknesses/questions pointed out below.
>
> 1. **Potentially unfair comparison (tokenizer / embedding optimizations for random models)**
> We acknowledge this concern and have added additional **new** results:
> - **Paired, ceteris‑paribus finetuning (Fig. 1; Sec. 3.1).** The main finetuning analysis now plots, for each architecture, _pretrained (y)_ vs. its _own_ randomly initialized counterpart (x) with identical tokenizer, width, positional encoding, and training schedule; the vertical gap is the pretraining gain for that architecture. This makes two patterns explicit: character tokenizers yield stronger random baselines; subword (k‑mer/BPE) models show larger gains from pretraining; the character‑tokenized Mistral still benefits, indicating an interaction with architecture/scale. (Fig. 1; NT/GUE/Genomic Benchmarks).
> - **Tokenizer ablation with matched architecture (Table 3; Sec. 3.3).** We pretrained two _identical_ HyenaDNA models differing only in tokenizer (char vs. k‑mer). Despite a _lower_ pretraining loss for the character model, the k‑mer model achieved a **+0.187** average downstream MCC advantage across tasks—direct evidence that tokenizer‑induced inductive bias can dominate downstream performance and that pretraining loss is a weak surrogate for discriminative genomics tasks.
>
> **On “what if pretrained also used char + larger embeddings?”** Our pool already contains pretrained character‑token models (Mistral, Caduceus, HyenaDNA), and the paired analysis shows their gains are either negative or smaller relative to subword models; Mistral is a positive exception, consistent with architecture × scale interactions. For feature extraction, increasing embedding dimension markedly improves _random_ models (e.g., HyenaDNA 0.690 -> 0.753 F1; NTv2‑50M 0.482->0.703 when switched to char + larger embeddings; Table 2) shows that representation quality, not pretraining per se, drives the zero‑shot utility. Additionally, Fig. 5 (Sec. 3.4) sweeps pretrained and random HyenaDNA across embedding dimensions and shows that pretraining helps only at d=64; from d ≥ 128 the random counterpart matches or exceeds the pretrained model even when architecture and width are identical.
>
> 2. **Analysis of pretraining (“design issues” vs. true effect)**
>
> Our new and existing results, with revised key messages and discussion now help separate the what from why.
>
> -   **What –**  Small randomly initialized character‑token models are often competitive with, or exceed, larger pretrained subword models across NT, GUE, and Genomic Benchmarks (e.g., random Caduceus surpasses NT‑500M by ~0.1 MCC on H3K9ac/H3K4me1/H3K36me3 despite 60× fewer parameters; Fig. 1 and Sec. 3.1).
>
> -   **Why –**  We posit that the vocabulary size and token granularity set the  _random baseline difficulty_. Character vocabularies (12 tokens) pose an easier from‑scratch representational problem; large subword vocabularies (4k–32k) pose a harder problem that pretraining can amortize – hence larger gains for k‑mer/BPE in the paired plot and in low‑label settings (below). The tokenizer ablation (Table 2) helps confirm this.
>
> 3. **Task distribution and long‑range dependencies**
> We intentionally limited our analysis to the union of NT, GUE, and Genomic Benchmarks (52 tasks covering structural, functional, and regulatory genomics tasks, and have been  widely used for GFMs). We do recognize the limitation with respect to the long-range tasks. Several included models have short context windows (128–512 tokens), which precludes a controlled comparison on 100k–200k bp tasks (gene‑expression regression, enhancer–promoter linkage). We now state this scope/limitation clearly in the Discussion and propose that the long‑context evaluations will be promising direction for future work with appropriate architectures.
>
> 4. **Generative models (Evo2, GenomeOcean)**
> Our claims are now clearly scoped to discriminative classification and frozen‑feature regimes. We do not assert conclusions for generative sequence design. The Discussion now names generative modeling as a distinct setting (objective, metrics) where pretraining may well be valuable; our results simply show that on  _standard, widely-used classification benchmarks_, tokenizer/architecture choices can dominate the measured effect of pretraining. We direct the readers towards these interesting new models in our Discussion.
>
> 5. **Quantized models (GERM)**
> We evaluate full‑precision checkpoints. Quantization introduces performance penalty; whether it helps or hurts is orthogonal to the central finding that, even at full precision, many GFMs lack allele‑level sensitivity and exhibit limited finetuning gains relative to strong random baselines. We therefore note quantization as potential future work in Discussion, and direct the interested readers towards GERM as an example.

---

> > ### Comment · Reviewer_rYMn · 2025-11-27
> >
> > Thank you for the detailed response and the additional experiments. These have addressed my concerns, and I am willing to raise my rating from 4 to 6.

---

### Official Review · Reviewer_4ABt · 2025-10-30

**Soundness:** 2
**Presentation:** 2
**Contribution:** 2
**Rating:** 4
**Confidence:** 4

**Summary:**

This paper challenges the prevailing assumption that large-scale unsupervised pretraining is inherently beneficial for Genomic Foundation Models (GFMs). The authors conduct an extensive empirical study comparing seven different GFMs against their randomly initialized counterparts across 52 diverse genomic tasks, spanning finetuning and feature extraction. The core finding is that randomly initialized models, when properly tuned, often match or even exceed the performance of their billion-scale pretrained counterparts.

Furthermore, the paper introduces a critical set of analyses on genomic variation, demonstrating that current GFMs are largely insensitive to subtle, clinically relevant mutations like SNPs. The models perform at near-random chance on variant classification tasks and produce nearly identical embeddings for reference and mutated sequences. The authors conclude that current pretraining strategies, largely adapted from NLP, are insufficient for genomics and that the field must rethink its approach to tokenization, pretraining objectives, and evaluation.

**Strengths:**

1. Large-Scale, Rigorous Evaluation: The sheer scale of the study (7 models, 52 tasks) provides very strong evidence. The authors' commitment to a rigorous hyperparameter search for all models (random and pretrained) makes their comparison fair and robust.

2. Novel and Critical Analysis: The genomic variation analysis (Sec 3.3) is a major strength. By showing that GFMs are insensitive to SNPs and fail on ClinVar data, the authors expose a critical blind spot in current models and evaluation methods. This finding alone is a significant contribution.

3. Actionable Insights: The paper provides clear hypotheses for these failures (k-mer tokenization obscuring SNPs, high masking rates) and points toward concrete areas for improvement (e.g., character-level tokenizers, as used in their own Mistral model).

**Weaknesses:**

1. Limited Scope (Generative Tasks): The paper's claims are based entirely on classification and feature extraction tasks. The authors briefly concede in the discussion and conclusion that pretraining might still be valuable for generative tasks. This is an important limitation, and the bold title ("Pretraining Does Not Promise Performance") might be a slight overstatement. The weakness is minor, as the paper's scope is already large, but it should be stated more prominently.

2. "What" over "Why": The paper excels at showing that pretraining fails but is less definitive on why. The discussion of tokenization and masking rates is insightful but largely correlational. The paper would be strengthened by even a single targeted ablation, e.g., pretraining two identical small models, one with k-mer and one with character-level tokenization, to prove that the tokenizer is the key factor.

**Questions:**

1. The authors convincingly argue that k-mer/BPE tokenizers are a major issue, especially for the genomic variation tasks. Their own Mistral model, which uses a character tokenizer and performs well, seems to support this. Could the paper's central finding be more narrowly (and perhaps more accurately) stated as "Current k-mer-based pretraining strategies are ineffective" rather than a blanket statement about all pretraining?

2. Following on the previous point, the feature extraction results (Table 2, Fig. 3) suggest that a randomly-initialized model with a character tokenizer and an optimized embedding dimension is a top performer. Is this also true for the finetuning tasks? A comparison of "best random (char-tokenizer)" vs. "best pretrained" would be very illuminating.

3. The authors attribute their different findings from prior work (e.g., Dalla-Torre et al., 2024) to their more rigorous hyperparameter search. As a sanity check, were they able to reproduce the original paper's results (i.e., showing a benefit for pretraining) by using the original fixed hyperparameters? This would definitively confirm that the HPO sweep is the key methodological difference.

---

> ### Author Response · Authors · 2025-11-27
> **Response to Reviewer – 4ABt**
>
> **Please also see the “COMMENT FOR ALL REVIEWERS” for a high‑level summary of changes.**
>
> We sincerely appreciate the thorough review and constructive framing. Below we address scope, mechanism (“what vs. why”), and specific questions.
>
> **Weakness 1. Limited Scope (Generative Tasks):**
> Our claims are now clearly scoped to discriminative classification and frozen‑feature settings. The paper evaluates 7 GFMs across 52 tasks (NT, GUE, Genomic Benchmarks) with nearly 10k finetuning runs and per‑model LR sweeps, plus a feature‑extraction regime (biotype / ancestry), and variant sensitivity (ClinVar LLR / cosine similarity) to probe representation quality. We now state this scope prominently in the Discussion and Conclusion, and acknowledge that the generative tasks are grounds for important future work.
>
> **Weakness 2. "What" over "Why"**
>
> We add two analyses that now directly isolate mechanism, and address the reviewer's ask for an experiment to assess this.
>
> 1.  **Paired, within‑architecture comparison (new Fig. 1, Sec. 3.1).**  Each point compares a model to  _itself_: pretrained (y) vs. its own random initialization (x), holding tokenizer, width, positional encoding, and schedule fixed. Reading left‑to‑right (random baseline) and bottom‑to‑top (pretrained) makes two patterns explicit: (i) character tokenization yields stronger random baselines; (ii) subword models (k‑mer/BPE) exhibit larger pretraining gains, whereas character models are smaller/variable (Mistral is a positive exception), indicating an interaction between tokenizer and architecture.
>
> 2. **Tokenizer ablation with matched architecture (new Table 3 / Sec. 3.3).** Two identical HyenaDNA models differ only in tokenizer (char vs k‑mer). Despite a _lower_ pretraining loss for the character model, the k‑mer model achieves **+0.187**average MCC across downstream tasks. This establishes **causal** evidence that tokenizer‑induced inductive bias can dominate downstream performance and that pretraining loss is a weak surrogate for discriminative genomics.
>
> These results support a revised key message – **pretraining gains are gated by tokenizer/architecture alignment**, not guaranteed by scale alone. That alignment is visible in the manuscript’s Table 1 (char vocab = 12 vs k‑mer = 4107 / BPE = 4096/32K), which helps explain the strong random baselines for character models and why pretraining amortizes a harder representation problem for large‑vocabulary subword models.
>
> **Q1 (narrower central finding).** We now state the finding as described above – subword tokenizers benefit consistently from pretraining; character tokenizers yield strong from‑scratch baselines, with pretraining benefit depending on architecture/scale (Mistral still gains). This framing replaces previous statements and is reflected in the Discussion/elsewhere in the paper.
>
> **Q2 (best random char vs best pretrained under finetuning).**  Yes, also true for the finetuning tasks. On NT histone tasks, the randomly initialized  **Caduceus**  (8 M params, char) frequently outperforms larger pretrained models, including NT‑500M, NTv2‑50M, and GENA‑LM (e.g., H3K9ac/H3K4me1/H3K36me3; often by ~0.1 MCC vs NT‑500M, despite 60× fewer parameters). Similar patterns appear in GUE (e.g., Core Promoter Detection where random Mistral tops all pretrained) and in Genomic Benchmarks. We show per‑task panels and subgroup summaries to make this explicit.
> Additionally, we show in fig. 5, for a matched experiment, increase in embedding dim doesn’t help HyenaDNA improve performance of its pretrained model over random.
>
> **Q3 (reproducing prior “pretraining helps” with fixed hyperparameters).** With fixed hyperparameters, we indeed reproduce a qualitative “pretraining advantage.” However, per‑model LR sweeps and full finetuning erase or shrink many gaps; LoRA underperforms full finetuning in our ablation (Table 8), so we base all primary comparisons on full finetuning under tuned LRs (Table 9 lists the sweep). These choices explain divergences from earlier reports and are now emphasized in the Methods/Discussion.
>
> Further comments
>
> **Variant sensitivity (summary result relevant to your strength 3 comment).** Across TP53/BRCA2/CFTR ClinVar, encoder and decoder GFMs yield **AUROC ≈ 0.35–0.54** under site‑wise LLR; cosine similarities between reference and mutated sequences are typically **≥ 0.98** even with mutation‑site pooling. These outcomes are reported in Tables 4–5 and Fig. 6, and we specify the pseudo‑LLR protocol for encoders in Methods. The observation aligns with your assessment that subword tokenization blurs SNV granularity; we also note that MLM masking at 15% further misaligns the pretraining objective with per‑base variation rates.
>
> **Low‑label regimes.**  We add learning‑curves at 1/5/10/50% labels (Fig. 3). Pretraining helps most for subword models (e.g., DNABERT‑2 gains ≳ +0.25 MCC on H3K4me1 at 1%), while character models see small/variable transfer. This reinforces the tokenizer‑gated view.

---

### Official Review · Reviewer_zXbv · 2025-11-01

**Soundness:** 2
**Presentation:** 3
**Contribution:** 2
**Rating:** 4
**Confidence:** 5

**Summary:**

This submission evaluates seven Genomic Foundation Models (GFMs) on 52 tasks spanning NT Benchmark, GUE, and Genomic Benchmarks, comparing pre-trained checkpoints against randomly initialized counterparts under fine-tuning, frozen‑feature (“feature extraction”), and genomic variation settings. The headline result is that models trained from scratch often match or surpass their pre-trained versions; moreover, embeddings from multiple GFMs are notably insensitive to clinically relevant variants, with cosine similarity remaining ~0.9–0.999 even after many SNPs are introduced. The paper argues that (i) existing NLP‑style pre-training is a poor investment for regulatory genomics classification and (ii) tokenization/architectural choices (e.g., character tokenization, larger embedding dimensions) dominate performance.

**Strengths:**

- **(S1)** Breadth and scale of evaluation with reproducibility intent. The study spans 7 models and 52 tasks across three popular benchmarks, with nearly 10k fine-tuning runs and LR sweeps, and reports model‑wise subgroup results. Datasets and checkpoints are standard, and code is linked (anonymized) in the paper, which, taken together, improves reproducibility. Meanwhile, the Figure 1 summary is easy to parse and consistently shows a limited advantage for pre-training when judged against a strong random baseline. The authors also empirically prefer full fine-tuning over LoRA, reducing a common confound in cross‑model comparisons.

- **(S2)** Variant-sensitivity diagnostics reveal a critical gap. The mutation-sensitivity experiments and ClinVar log-likelihood–ratio AUROCs near 0.5 jointly show that many GFMs’ representations are remarkably insensitive to clinically relevant point mutations, echoing long-standing concerns that subword/k‑mer tokenization blurs SNV granularity.

- **(S3)** Positioning within ongoing benchmarking discussions. The paper’s thesis aligns with emerging benchmark evidence that SFT matters and that supervised long-range models can still dominate on gene expression, underscoring that what we benchmark matters as much as how.

**Weaknesses:**

- **(W1)** “Pre-training vs. random” comparisons are confounded by architecture/tokenizer changes. The feature-extraction claim that random models can beat pre-trained hinges on changing the tokenizer (char rather than original k‑mer/BPE) and increasing the embedding size for the random arm (Table 2 and Figure 3), whereas pre-trained arms keep their native tokenizers/widths. This violates ceteris paribus, conflating the benefit of model design with the absence of pre-training. A fair test needs identical architectures/tokenizers/widths with/without pre-training. As is, the headline conclusion is directionally plausible but not causally isolated.

- **(W2)** Variant sensitivity method is too blunt to support strong claims. The mutation‑sensitivity analysis relies on global pooling and cosine similarity of full‑sequence embeddings, while high similarities that sometimes increase with more mutations likely reflect pooling/normalization effects rather than true biological blindness. Moreover, the LLR analysis lacks detail for encoder-only masked LMs (pseudo-likelihood vs. left-to-right), complicating an apples-to-apples comparison with decoders. Token-level distances, attribution at mutated loci, and clearly specified pseudo-LLR for encoders are necessary.

- **(W3)** Task coverage underweights where long-range biology is known to matter. The selected tasks span mainstream benchmarks, but the suite lacks gene-expression regression (e.g., bulk RNA/CAGE) and enhancer–gene linkage tasks for which long-range inductive biases and explicit supervised training (e.g., Enformer) remain strong baselines and stress tests.

- **(W4)** Positioning more concurrent architecture findings could be sharper. The paper’s narrative at times attributes the observed wins to the absence of pretraining, when the architecture/inductive bias could be dominant. Independent recent results indicate that simple, well-tuned CNNs can outperform SSM/Transformer DNA models on many tasks without pretraining (e.g., ConvNova), suggesting that architectural priors and receptive-field design can rival or exceed pretraining gains. This alternative explanation deserves explicit treatment in the Discussion.

**Questions:**

- **(Q1)** Can the authors provide apple-to-apple ablations where architecture, tokenizer, embedding size, positional encoding, and training schedule are identical, and the only difference is with vs. without pre-training? This would directly test the causal value of pre-training beyond the confounds noted in Table 2 and Figure 3.

- **(Q2)** How do the findings change in label-scarce regimes (e.g., 1%, 5%, and 10% of labels)? Please include learning curves and area under the data curve statistics to test whether pre-training is more helpful at low data.

- **(Q3)** For mutation sensitivity, could the authors further provide in-silico mutagenesis with base-resolution attribution (e.g., Grad-CAM) and compare to CADD/Enformer scores and to eQTL/sQTL ground truths (AUPRC, per-gene AUROC)?

**Details Of Ethics Concerns:**

There are no obvious Code of Ethics concerns. The work uses public genomic resources (IGSR/1000G, GRCh38, GENCODE, ClinVar) and reports only aggregate performance. Please confirm that all data is de‑identified and used under the respective licenses.

---

> ### Author Response · Authors · 2025-11-27
> **Response to Reviewer – zXbv**
>
> **Please also see the “COMMENT FOR ALL REVIEWERS” for a high‑level summary of changes.**
>
> **W1 / Q1 / W4 — Architectural/tokenizer confounds; ceteris paribus; inductive bias**
> We agree that the original comparison conflated architectural choices with the effect of pretraining. We address this in 3 ways for more accurate messaging in the paper.
>
> 1. **Paired random vs. pretrained (Fig. 1; Fig. 5, Sec. 3.1, 3.4).**
> We thoroughly extended our main finetuning analysis, for each architecture we plot pretrained (y) vs. its own random init (x) with identical tokenizer, width, positional encoding, and schedule. Above‑diagonal = pretraining gain. Key readouts:
> 	- Character‑token models (Caduceus, HyenaDNA, Mistral) have higher random baselines than k‑mer/BPE.
> 	- Pretraining yields larger gains for k‑mer/BPE; gains for character models are smaller/variable (Mistral an exception).
> 	- Small random models (e.g., Caduceus) often match/beat larger pretrained GFMs across NT, GUE, and Genomic Benchmarks.
> - For a matched experiment (fig. 5), increase in embedding dim doesn’t help HyenaDNA improve performance of its pretrained model over random.
> 2. **Tokenizer ablation with matched architectures (Table 3; Sec. 3.3).**
> To go beyond correlational evidence, we implemented the targeted ablation you suggested. With two HyenaDNA models, identical except tokenizer, are pretrained and evaluated:  Table 3 shows that:
> 	-   The character model has lower pretraining loss (1.18 vs 1.215).
> 	-  The k‑mer model nonetheless achieves an average downstream MCC improvement of ~0.19, and performs better on each individual task.
> 	- **Takeaway –** tokenizer‑induced inductive bias can dominate downstream performance; pretraining loss is a poor proxy for discriminative genomics tasks.
>
> 3. **Discussion reframing.**
>  In response to W4, we rewrote the Discussion to frame our findings as tokenizer‑ and architecture‑dependent rather than as a blanket failure of pretraining. We made changes to abstract, intro and other sections to align this key messaging throughout the revised paper.
>
> These changes provide the requested comparison and clarify the scope.
>
> ---
> **Q2 — Label‑scarce regimes (Sec. 3.2; Fig. 3).**
> - New low‑resource experiment in Sec. 3.2 / Fig. 3.
> - Finetuned pretrained + random versions of 4 models, NT‑50M (k‑mer), DNABERT‑2 / GENA‑LM (BPE), HyenaDNA (character), on four representative tasks from NT benchmark, with 1/5/10/50% of the labels.
> Findings
> 	- BPE (DNABERT‑2, GENA‑LM): consistent gains from pretraining across all fractions; largest at 1–5% (DNABERT‑2 ≳ +0.25 MCC on H3K4me1 at 1%).
> 	- Char (HyenaDNA): small/neg. gains in the same low‑label settings.
> 	- k‑mer (NT‑50M): intermediate.
> - **Conclusion–** tokenizer‑gated pattern strengthens with fewer labels; pretraining helps when subword vocabularies make from‑scratch learning harder.
>
> ---
>
> **W2 / Q3 — Variant‑sensitivity methodology & pseudo‑LLR (Sec. 3.5).**
> With variant analyses, we wanted to test a basic question – do current GFMs change their representations meaningfully in the presence of clinically annotated SNPs, without finetuning? We agree that clarifying the methodology is important. In the revision we
> -   Explicitly state that we use two pooling strategies in the mutation‑sensitivity analysis (Sec. 3.3)
> 	1.  global last/CLS pooling; and
> 	2.  mutation‑site pooling, which aggregates representations only at tokens overlapping the introduced SNPs. Mutation‑site pooling directly targets local changes and does not average over non‑mutated positions.
>
> -   Clarify the pseudo‑LLR computation for encoders – for masked LMs, we follow Benegas et al. (2025a) and approximate the log‑likelihood at the variant position using the softmax probabilities at a masked token, holding the surrounding context fixed. This brings encoder models in line with decoder‑style likelihoods.
>
> **Results.** Under both pooling schemes and pseudo‑LLR, cosine similarities for ref vs. mutated sequences typically ≥ 0.98; ClinVar AUROCs ≈ 0.35–0.54 across BRCA2/CFTR/TP53 (Tables 4–5; Fig. 6). Character‑token models show slightly lower similarities or stronger ancestry F1, but effect sizes remain modest.
> **Interpretation.** Current objectives/tokenizers do not reliably encode allele‑level signals. We have added details to methods; and noted that the token‑level distances and attribution vs. CADD/Enformer/eQTL/sQTL could be good future work in discussion.
>
> ---
>
> **W3 — Task coverage and long‑range biology.**
> Our suite is the union of NT, GUE, and Genomic Benchmarks (52 classification tasks commonly used for GFMs). Several evaluated models have short contexts (128–512 tokens), limiting controlled inclusion of 100k–200k bp tasks. We frame long‑range gene‑expression/enhancer–gene linkage as complementary future evaluations with long‑context architectures. The revised Discussion now states this limitation explicitly and points to these important, complementary future evaluations.

---

> > ### Comment · Reviewer_zXbv · 2025-11-27
> > **Feedback to Rebuttal**
> >
> > Thanks to the authors for their detailed responses with the comprehensive revision. After reading the responses, I found that some of my concerns were nearly addressed. Despite the improvements, some key issues remain only partially addressed:
> >
> > * The task scope limitation persists. The authors did not add any long-range genomic tasks, e.g., enhancer–promoter interaction prediction or gene expression regression, which I had highlighted as important scenarios where pretraining might prove beneficial. They acknowledge this gap in the discussion, which is fair, but it means the study’s conclusions still do not encompass long-range or truly genomic “language” tasks. Similarly, while the new low-data experiments help, the paper still lacks an explicit analysis of compute cost-effectiveness or scaling of pretraining vs. direct training. The authors assert that pretraining’s modest gains may not justify its high cost, but we still do not see a detailed stratification or experiment varying the compute budget to support that claim.
> >
> > * From a methodological novelty standpoint, the work remains largely a comprehensive empirical study rather than proposing new modeling techniques. The revised manuscript does improve its framing. However, the contributions of this work lie in rigorous evaluation and analysis. While these contributions are valuable, they fall on the empirical side. This limits the paper’s appeal for a venue like ICLR, where one expects a certain level of methodological or theoretical innovation. The authors’ insights about tokenizer choice, architecture, and task alignment are interesting, but they are presented as observations and post-hoc analyses rather than a new solution or model for the community to adopt.
> >
> > Overall, I believe this work is scientifically sound and useful, but not yet aligned with ICLR’s expectations for novelty and methodological contribution. I therefore maintain my original score and encourage the authors to consider a genomics or bioinformatics venue, where comprehensive, carefully controlled empirical studies.

---

> > > ### Author Response · Authors · 2025-12-03
> > > **Additional scaling experiments & answers**
> > >
> > > Dear Reviewer **zXbv**,
> > >
> > > Thank you for your continued engagement with our work. Below we address your outstanding comments:
> > >
> > > > **Q.** The paper still lacks an explicit analysis of compute cost-effectiveness or scaling of pretraining vs. direct training.
> > >
> > > We provide additional experiments for the scaling analysis you requested.
> > >
> > > In this new experiment, we analyzed how model size affects downstream performance for both the Nucleotide Transformer and HyenaDNA. For each model we've used 3 scales:
> > >
> > > - **NT-v2**:NT-v2-50M (\~56M params), NT-v2-100M (\~98M params), and NT-v2-250M (\~250M params).
> > > - **HyenaDNA**: HyenaDNA-tiny (451K params), HyenaDNA-medium (14.2M params), and HyenaDNA-large (54.6M params).
> > >
> > > For each model at each scale, we performed a learning rate sweep to ensure fair comparison between pretrained and random. Hyena models were trained for 100 epochs and NT models were trained for 30 epochs on two NT tasks: **enhancers** and **H3K4me1**. We report the best test MCC achieved across the LR sweep for each setting.
> > >
> > > *Table 1: HyenaDNA scaling*
> > >
> > > | Model | Params | Enhancers (Pre) | Enhancers (Rand) | Δ P - R | H3K4me1 (Pre) | H3K4me1 (Rand) | Δ P - R|
> > > |-------|--------|-----------------|------------------|---|---------------|----------------|---|
> > > | HyenaDNA-tiny | 451K | **0.370** | 0.329 | +0.041 | **0.334** | 0.324 | +0.010 |
> > > | HyenaDNA-medium | 14.2M | **0.245** | 0.210 | +0.035 | **0.220** | 0.205 | +0.015 |
> > > | HyenaDNA-large | 54.6M | **0.224** | 0.201 | +0.023 | 0.139 | **0.182** | -0.043 |
> > >
> > > *Table 2: NT scaling*
> > >
> > > | Model | Params | Enhancers (Pre) | Enhancers (Rand) | Δ P - R | H3K4me1 (Pre) | H3K4me1 (Rand) | Δ P - R |
> > > |-------|--------|-----------------|------------------|---|---------------|----------------|---|
> > > | NT-v2-50M | ~56M | **0.527** | 0.493 | +0.034 | 0.500 | **0.502** | -0.002 |
> > > | NT-v2-100M | ~98M | **0.548** | 0.505 | +0.043 | 0.350 | **0.500** | -0.150 |
> > > | NT-v2-250M | ~250M | 0.368 | **0.500** | -0.132 | 0.366 | **0.498** | -0.132 |
> > >
> > > **Observations:**
> > >
> > > 1. **Random models are competitive on every scale**:
> > >     * **NT**: random initialization beats pretrained on each scale for H3K4me1 and on the largest scale for enhancer tasks. For two smallest scales for enhancer tasks the advantage of pretrained is tiny.
> > >     * **HyenaDNA**: random initialization provides a very close performance to the pretrained version across all 3 scales.
> > >
> > > 2. **Inverse scaling phenomenon**: for both models smaller models perform better. This suggests these tasks don't benefit from increased model capacity.
> > >
> > > These results strengthen our central conclusion that the effectiveness of pretraining is not universal but is modulated by model scale, tokenizer choice, and task characteristics. The fragility of pretraining benefits at larger scales suggests that simply scaling up current GFM architectures is not a reliable path to improved performance.

---

> > > ### Author Response · Authors · 2025-12-03
> > > **Addressing methodological novelty**
> > >
> > > > **Q.** From a methodological novelty standpoint, the work remains largely a comprehensive empirical study rather than proposing new modeling techniques. While these contributions are valuable, they fall on the empirical side. This limits the paper's appeal for a venue like ICLR, where one expects a certain level of methodological or theoretical innovation.
> > >
> > > We respectfully disagree with this characterization. We would like to highlight several points:
> > >
> > > 1. **ICLR explicitly welcomes this type of contribution**: The ICLR Call for Papers (https://iclr.cc/Conferences/2026/CallForPapers) explicitly lists several topic areas that our work directly fits in:
> > >    - **"representation learning for computer vision, audio, language, and other modalities"** -- our work evaluates the quality of learned representations in genomic foundation models
> > >    - **"visualization or interpretation of learned representations"** -- we analyze what these models actually learn and whether pretraining produces meaningful representations
> > >    - **"datasets and benchmarks"** -- we provide a comprehensive benchmark evaluation across 52 tasks and introduce a novel ancestry classification benchmark for evaluating variant sensitivity
> > >
> > >    Our central contribution is understanding the quality of representations learned by GFMs: do they capture meaningful genomic patterns, or are they no better than random projections? This is fundamentally a representation learning question.
> > >
> > > 2. **Critical assessments with surprising findings drive progress**: science advances not only through new methods but also through rigorous evaluation that challenges prevailing assumptions. Our finding that pretraining benefits are fragile and tokenizer-dependent is surprising and counterintuitive to the genomics ML community. Such critical assessments prevent wasted computational resources and redirect research toward more productive directions.
> > >
> > > 3. **Substantial computational investment and rigor**: this work represents over **10,000 fine-tuning experiments** across 7 models and 52 tasks, with comprehensive hyperparameter sweeps. The scale and methodological rigor of this evaluation far exceeds typical empirical studies.
> > >
> > > 4. **Actionable insights**: we don't just identify problems, we provide concrete practical guidance in our Discussion (Outlook part):
> > >    - **(i) Short-range classification**: small character-tokenized models are strong and compute-efficient baselines; we recommend reporting well-tuned random baselines as standard practice
> > >    - **(ii) Subword models and label-scarce regimes**: practitioners should expect gains from pretraining, but must align the objective with use case (e.g., mutation-aware masking, contrastive signals at variant loci)
> > >    - **(iii) Variant-centric applications**: clear evidence that the community should prioritize tokenizer and objective function redesign before further scaling
> > >
> > >    Our tokenizer ablation (Table 3) establishes a causal mechanism, and these findings directly inform practitioners on model selection, tokenizer choice, and when pretraining is worth the computational investment.
> > >
> > > We believe our work makes a significant contribution by providing the most comprehensive evaluation of GFM pretraining to date, with findings that will directly impact how the community develops and deploys these models.

---

### Author Response · Authors · 2025-11-27
**COMMENT FOR ALL REVIEWERS**

We thank all reviewers for their thoughtful feedback. In response, we made three main technical additions and several clarifications.

1.  Paired comparison of each pretrained GFM to its own randomly initialized counterpart (new Fig. 1 and revised Sec. 3.1), and additional embedding dimension experiment (fig. 5), which directly addresses ceteris paribus concerns.

2.  Low‑resource finetuning experiments on NT histone and enhancer tasks at 1%, 5%, 10%, and 50% label fractions (new Fig. 3 and added paragraph in Sec. 3.2).

3.  A targeted tokenizer ablation with matched HyenaDNA architectures that differ only by tokenizer (character vs k‑mer), showing a ~0.19 average MCC gap despite a lower pretraining loss for the character model (new Table 3 and added paragraph in Sec. 3.3).


We also:

-   Moderated several statements to better reflect the more subtle picture that emerged.

-   Clarified our mutation‑sensitivity methodology (mutation‑site pooling, encoder pseudo‑LLR).

-   Explicitly scoped our conclusions to regulatory classification and feature extraction, while acknowledging long‑range and generative settings as important but out‑of‑scope.

-   Revised paper title.

Additionally to new Fig.1, we also highlight that Fig. 7–10 and Table 18 in the Appendix present a side-by-side comparison of pretrained vs random models.

We highlight the revised text in the updated manuscript in blue color.

## **Below we provide a summary of new analyses (highlights).**

### Paired pretrained‑vs‑random comparisons (Fig. 1, Sec. 3.1).
 - Plotting each model’s pretrained score against its own randomly
   initialized counterpart isolates the contribution of pretraining.
  - Character‑tokenized models start from stronger random baselines,
   whereas subword models gain more from pretraining.
  - Concretely, the k‑mer/BPE models show sizeable vertical improvements – e.g., NT‑500M +0.242 MCC and DNABERT‑2 +0.139 on GUE – while character models are variable (Caduceus −0.114 on GUE; HyenaDNA ≈0), and Mistral (character, larger/modern recipe) still benefits (+0.142on GUE).
  - These paired points make clear both the baseline ordering
   (left‑to‑right) and the pretraining benefit (bottom‑to‑top).

### Low‑data finetuning (Fig. 3, Sec. 3.2).
- At 1–10% labels, pretraining is most helpful when the tokenizer makes the from‑scratch learning problem harder – DNABERT‑2 exceeds its random counterpart by ~0.25 MCC at 1% on H3K4me1, and GENA‑LM shows consistent gaps across H3K4me1/3 and H3K9ac.
- In contrast, the character model HyenaDNA exhibits negligible or negative transfer in the same settings, and NTv2‑50M gains are smaller.

### Causal tokenizer ablation (Table 3, Sec. 3.3).
- We pretrain two identical HyenaDNA architectures that differ only in tokenizer. Despite the character model achieving a lower pretraining loss (1.180 vs 1.215), the k‑mer model delivers +0.187 average MCC across H3K4me3/H3K9ac/Enhancers.
- This isolates tokenizer inductive bias as one primary driver of downstream performance and shows that pretraining loss is a poor proxy for discriminative genomics tasks; a complementary HyenaDNA embed dim sweep (Fig. 5, Sec. 3.4) further shows that even with matched architecture and width, pretraining only helps at d=64 and is matched or exceeded by random initializations at larger dimensions typically used.

### Variant sensitivity clarifications (Sec. 3.5).
- Using mutation‑site pooling (addressing the concerns with global pooling) and a site‑wise LLR protocol for encoder LMs, embedding similarities between reference and mutated sequences remain high (often ≥ 0.95), and ClinVar AUROCs are near random (e.g., 0.345–0.536).
- Strong evidence for insufficient allele‑level sensitivity in current GFMs (see Fig. 6; Tables 4–5).

Collectively, these additions bring us to consistent message – the utility of pretraining in regulatory classification is gated by tokenizer/architecture alignment. Subword models benefit most from pretraining; character models provide strong random baselines and show heterogeneous gains unless paired with stronger architectures; and variant‑level tasks remain unsolved by current objectives. Full plots, tables, and implementation details are in the revision (Fig. 1, Fig. 3, Fig. 5, Table 2; Secs. 3.1–3.4). Full paper has been revised (including Abstract, Introduction, Results, Discussion, Conclusion) to ensure the key messages are aligned and reflect the updated results.

We sincerely thank the reviewers for helping us significantly strengthen the paper.

---

### Meta-Review · Area_Chair_JGVw · 2025-12-29

**Summary:**

The paper attempts to conduct a thorough analysis of whether pretraining of genomic foundation models actually contributes to downstream task performance under three conditions 1) fine-tuning for functional element prediction, 2) extracting features from a frozen model and then training a classifier on top for biotype prediction, and 3) addition mutations and then extracting features from a frozen model to perform mutation sensitivity analysis and ancestry prediction.  The conclusion of the work is that for some models (Caduceus, HyenaDNA) pretraining does not always help, and an randomly initialized model can perform as well.  In the revision, the authors also added additional experiments showing the important of the tokenizer with the performance of the randomly initialized model strongly correlating with the tokenizer.

The AC finds the experimental results from the work to be interesting.  However the AC also note that the work is missing important work that was not discussed including: [1] prior work that investigates impact of different training stratigies for gFMs (including random initialization and pretrained) and [2] impact of tokenization on performance.  The AC believes there is still value in the work, but recommends the authors add discussion of these relevant work.

Missing references

[1] No Clear Winner at Small Scale: Comparing Modern Sequence Architectures and Training Strategies for Genomic Language Models [Milovanović and Orvieto. Generative AI for Biology Workshop at ICML 2025]

[2] Model Decides How to Tokenize: Adaptive DNASequence Tokenization with MxDNA [Qiao et al. NeurIPS 2024]

**Reviewer Concerns:**

Reviewers expressed the following concerns:
1. Confounding factors in experiments [zXbv, rYMn], need more targeted ablations [4ABt]
   - *Mostly addressed.* Additional experiments were added 1) directly comparing pretrained model with randomly initializer variant (Fig 1) with tokenzier clearly indicated, 2) comparing impact of tokenizer (Tab 3), 3) pretraining with different amounts of data (Fig 3) and different embeddings sizes (Fig 5 for HyenaDNA)
2. Variant sensitivity experiments should be able to identify local mutations vs just looking at full sequence similarities [zXbv]
   - *Addressed.* Discussion added (out-of-scope of current work)
3. Limited scope of tasks [zXbv, 4ABt, rYMn]
   1. Does not cover long-range genomic tasks (e.g. enhancer–promoter interaction prediction or gene expression regression) [zXbv, rYMn]
   2. Missing generative tasks [4ABt, rYMn]
   - *Addressed.* Discussion added (out-of-scope of current work)
4. Question whether findings hold for quantized models [rYMn]
   - *Addressed.* Discussion added (out-of-scope of current work)
5. Empirical study versus providing a new modeling technique [zXbv]
   - *Addressed.* As the authors point out, empirical assessment of learned representations are valuable to the community.

**Reviewer Scores:**

The work received marginal reject from all reviewers (zXbv, 4ABt, rYMn).  On Nov 27th, the authors provided an author response and updated manuscript with clearly marked revisions.

Reviewer zXbv indicated that their concern was partially addressed but wishes to maintain their score.  Reviewer rYMn indicated willingness to increase their score to 6.  Reviewer 4ABt did not engage in discussion.

---

### Decision · Program_Chairs · 2026-01-26

Accept (Poster)